# High-resolution view of HIV-1 reverse transcriptase initiation complexes and inhibition by NNRTI drugs

Betty Ha [1,2,7], Kevin P. Larsen [1,3,5,7], Jingji Zhang[1,7], Ziao Fu[4,6], Elizabeth Montabana[1], Lynnette N. Jackson [1], Dong-Hua Chen [1] & Elisabetta Viani Puglisi [1✉]

Reverse transcription of the HIV-1 viral RNA genome (vRNA) is an integral step in virus replication. Upon viral entry, HIV-1 reverse transcriptase (RT) initiates from a host tRNA$^{Lys}_3$ primer bound to the vRNA genome and is the target of key antivirals, such as non-nucleoside reverse transcriptase inhibitors (NNRTIs). Initiation proceeds slowly with discrete pausing events along the vRNA template. Despite prior medium-resolution structural characterization of reverse transcriptase initiation complexes (RTICs), higher-resolution structures of the RTIC are needed to understand the molecular mechanisms that underlie initiation. Here we report cryo-EM structures of the core RTIC, RTIC–nevirapine, and RTIC–efavirenz complexes at 2.8, 3.1, and 2.9 Å, respectively. In combination with biochemical studies, these data suggest a basis for rapid dissociation kinetics of RT from the vRNA–tRNA$^{Lys}_3$ initiation complex and reveal a specific structural mechanism of nucleic acid conformational stabilization during initiation. Finally, our results show that NNRTIs inhibit the RTIC and exacerbate discrete pausing during early reverse transcription.

---

[1] Department of Structural Biology, Stanford University School of Medicine, Stanford, CA, USA. [2] Department of Molecular and Cellular Physiology, Stanford University School of Medicine, Stanford, CA, USA. [3] Program in Biophysics, Stanford University, Stanford, CA, USA. [4] Department of Chemistry and Molecular Biophysics, Columbia University, New York, NY, USA. [5] Present address: Department of Molecular and Cell Biology, University of California Berkeley, Berkeley, CA, USA. [6] Present address: Laboratory of Molecular Neurobiology and Biophysics, The Rockefeller University, Howard Hughes Medical Institute, New York, NY, USA. [7] These authors contributed equally: Betty Ha, Kevin P. Larsen, Jingji Zhang. ✉email: epuglisi@stanford.edu

Reverse transcription of the HIV-1 single-stranded (ss) RNA genome into double-stranded (ds) DNA is an essential early step in viral replication and a major target for current antiretroviral therapies[1]. A packaged viral enzyme, reverse transcriptase (RT), initiates DNA synthesis at the 3'-end of a host tRNA$^{Lys}_3$ that is part of a binary complex preassembled with the 5'-end of the HIV-1 viral RNA, called the primer-binding site (PBS). Kinetic investigations have shown that reverse transcription initiation on this vRNA–tRNA$^{Lys}_3$ template–primer complex is slow (~500-fold slower than elongation), nonprocessive, and displays a distinct pausing pattern along with the viral genomic RNA (vRNA) template[2–6]. By contrast, the elongation phase of reverse transcription, in which RT associates with DNA–DNA and DNA–RNA substrates, occurs with rapid and processive polymerase kinetics and has been well-characterized by a wide range of structural, biophysical, and biochemical approaches. RT in complex with nucleic acid substrates (DNA–DNA and DNA–RNA duplexes) has been captured in multiple structural states representative of the RT elongation polymerase catalytic cycle and RNase H engagement[1,7].

In contrast, RT initiation has been challenging to investigate using structural methods[5,8–10]. By encompassing the essential conserved features within the tRNA$^{Lys}_3$ primer and 101 nucleotides within the vRNA template that modulate RT initiation in vitro and in vivo, cryo-EM and X-ray crystallographic investigations have provided the first structural glimpses of how RT recognizes vRNA–tRNA$^{Lys}_3$ template–primer complexes[11–14]. A cryo-EM reconstruction revealed the overall architecture of the complex[12]. The 18 base-pair PBS helix sits in the RT cleft, and is extended by an additional four base pairs between vRNA and tRNA. The tRNA 5'-end refolds to form a coaxially stacked, long helical structure protruding from the RNase H domain of RT. The vRNA forms two helical regions H1 and H2 above the RT polymerase-active site, and a single-stranded connecting loop that bridges from RNA located in the RNase H domain to that near the RT polymerase domain. The structures of the core region of the RTIC revealed features that could explain the slow nature of initiation: RT adopts an open conformation with a hyperextended thumb, and the tRNA primer terminus is deviated away from the active site relative to structures representative of elongation[12–14]. However, the resolution of these structures, limited to 4.1 Å (cryo-EM, core)[13] and 3.95 Å (X-ray crystallography)[14], has prevented accurate mapping of protein–RNA contacts, and modeling of ions and small-molecule ligands, which is required to understand drug binding.

Reverse transcription is a key target for antivirals, including nucleoside analogs that act as chain terminators and non-nucleoside inhibitors (NNRTIs) that allosterically disrupt enzyme function[7]. Two of the most well-characterized NNRTIs include nevirapine (NVP), a first-generation inhibitor, and efavirenz (EFZ), a more efficacious and higher-affinity compound. Structures of RT in complex with elongation substrates and NNRTIs have uncovered the mechanism by which these drugs inhibit the chemistry step of polymerization, by allosteric conformational changes that destabilize the RT–nucleic acid complex[7,15]. NNRTI binding distorts the active site, repositions the primer grip, displaces the 3'-primer terminus, and loosens the thumb and finger clamp[15–18]. NNRTIs can also increase RNase H activity when RT is bound to an RNA–DNA elongation intermediate[7]. NNRTIs substantially reduce the buildup of early reverse transcription products, implying that NNRTIs may also impede initiation[19]. Notably, the recently determined low-resolution structures of the RTIC all exhibit a partially open RT-NNRTI-binding pocket, a feature previously absent in all RT structures lacking a bound NNRTI[12–14]. In addition, the complex studied by cryo-EM was also shown to be inhibited by NVP in biochemical assays[12].

However, whether or how NNRTIs perturb the conformation of an RTIC remains unknown. The slow nature of initiation suggests an inherent vulnerability of the RTIC to disruption, making it a particularly appealing target for the discovery of antivirals.

Here, we apply a combined biochemical and structural approach to delineate the protein–RNA contacts within the RTIC and determine the origins of NNRTI action on initiation. Using a minimal, active RNA initiation complex that spans the portions of vRNA and tRNA that reside within the RT cleft, while eliminating dynamic peripheral RNA elements that do not interact with the protein (Fig. 1a), we solved the structure of the minimal RT–vRNA–tRNA$^{Lys}_3$ initiation complex (miniRTIC) to 2.8-Å resolution, allowing for de novo modeling and definition of RT–RNA contacts. Using a similar approach, we also determined structures of the miniRTIC–NNRTI complexes with NVP and EFZ (3.1- and 2.9-Å resolution, respectively). The structural and biochemical data confirm the potent inhibitory activity of these drugs against initiation. Our results provide a high-resolution view of the RTIC and suggest how drugs might selectively target initiation.

## Results

**Construct design and validation.** Guided by prior structural and biophysical studies, we designed a minimal RTIC (miniRTIC) system that maintains the protein–dsRNA core while eliminating regions of segmental flexibility, which included peripheral elements of the vRNA (H1, H2, and connecting loop) and tRNA[12–14]. The resulting minimal vRNA–tRNA template–primer complex contained a 26-nucleotide fragment of the vRNA genome and a 39-nucleotide truncated tRNA$^{Lys}_3$. This bimolecular complex encompasses 4 nt of single-stranded template vRNA, the 22-bp extended PBS helix, and 6 bp of the tRNA primer helix capped by a GNRA loop (Fig. 1a). To circumvent the rapid dissociation of RT from dsRNA substrates, we adapted the previously employed disulfide cross-linking system to covalently link the RT p66 thumb domain to the vRNA–tRNA template–primer complex[12,13,20] (Supplementary Fig. 1b). After performing a 24-h cross-linking reaction, unbound RT and vRNA–tRNA template–primer complex were removed using multiple chromatography steps to generate the final pure miniRTIC ternary complex (Fig. 1b, c). We assessed the activity of the miniRTIC and found that it incorporates the next templated dNTP at a rate ($k_{pol}$) ~threefold faster than an equivalent un-cross-linked complex (Fig. 1d), and similar to what was observed with the larger (101nt)vRNA–(76nt)tRNA$^{Lys}_3$ template–primer complex studied previously (miniRTIC $k_{slow} = 0.002617\ s^{-1}$, +1 RTIC $k_{slow} = 0.003140\ s^{-1}$)[12]. The addition of NNRTIs substantially reduced the rate of nucleotide incorporation by the miniRTIC: EFZ slowed incorporation by ~20-fold, whereas NVP slowed incorporation by ~16-fold (Fig. 1d and Supplementary Fig. 1e). Together, these data confirm that the miniRTIC represents an active functional state appropriate for the study of early reverse transcription initiation (i.e., prior to the incorporation of templated dNTPs) and its inhibition by antiretrovirals.

**Structure of the apo-miniRTIC.** We next optimized our prior cryo-EM freezing conditions to achieve monodisperse single particles in thin ice, a necessary prerequisite for higher-resolution single-particle cryo-EM[12,13]. The reconstruction of the ~138-kDa miniRTIC was resolved at an overall resolution of 2.8 Å, an ~1.3-Å improvement in the resolution from our previous +1 RTIC core structure[13] (Fig. 2a, Supplementary Figs. 2a–d and 3b, e, and Table 1). All RT p66 and p51 subdomains were readily modeled into the density using prior RT structures as a guide. Unlike our prior RTIC cryo-EM structures, the majority of protein

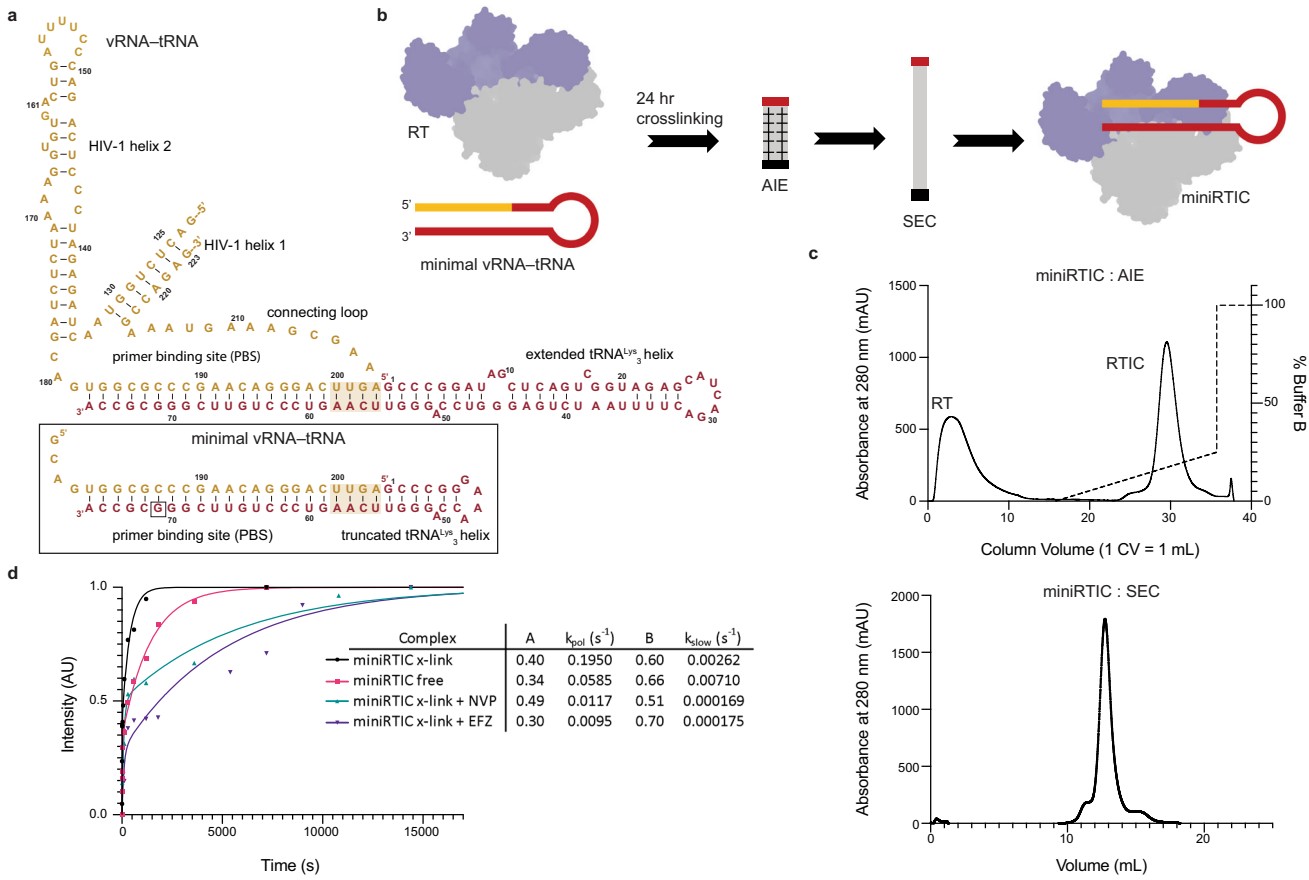

**Fig. 1 RTIC construct design, purification, and validation. a** Secondary structure of the vRNA–tRNA (gold-red) from Larsen et al.[12] (upper). The minimal bimolecular vRNA–tRNA template–primer (gold-red) construct (lower, boxed) was designed to encompass 4 nucleotides of single-stranded template vRNA, 22-bp of extended PBS helix, and 6 bp of tRNA primer capped with a GNRA loop; flexible RNA regions in the vRNA–tRNA from Larsen et al.[12] were excluded from the minimal vRNA–tRNA. **b** The miniRTIC (p66 subunit in purple; p51 subunit in gray) was formed by cross-linking RT and the minimal vRNA–tRNA for 24 h and purified by subsequent anion-exchange (AIE) and size-exclusion chromatography (SEC). **c** AIE and SEC chromatograms of the miniRTIC purification process. **d** Functional analysis of cross-linked (x-link) and un-cross-linked (free) miniRTIC. Incorporation assays of free miniRTIC (pink), x-link miniRTIC (black), x-link miniRTIC with 50 nM nevirapine (NVP, green), and x-link miniRTIC with 50 nM efavirenz (EFZ, purple) were initiated by addition of $\alpha$-$^{32}$P-dCTP and quenched at different time points. Data were fit using the relationship: Intensity = $A(1 - e^{-k_{pol}t}) + B(1 - e^{-k_{slow}t})$, where $A$ and $B$ represent the amplitude of the fast and slow processes, respectively, $k_{pol}$ is the apparent extension rate constant, and $k_{slow}$ is the rate of the slow process. $k_{pol}$ is ~3.3-fold faster for the x-link miniRTIC than for the free miniRTIC. Assays for the x-link miniRTIC and free miniRTIC were each repeated five times. NVP and EFZ conditions were each repeated three times to ensure reproducibility.

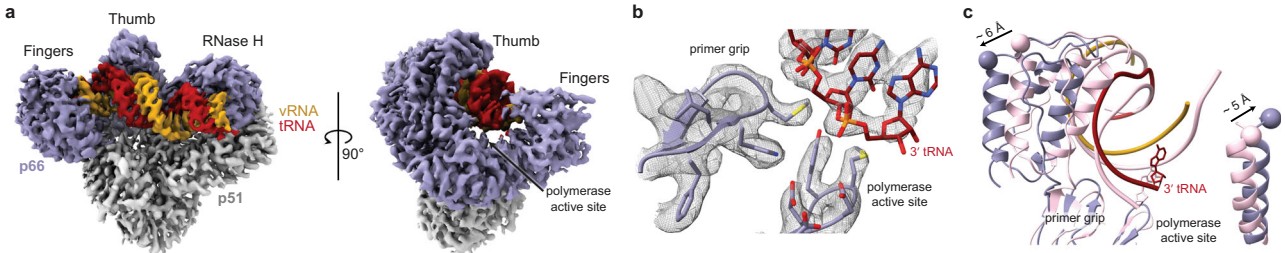

**Fig. 2 Improved resolution of the HIV-1 RTIC structure. a** In all, 2.8-Å map of the miniRTIC colored by chain (p66 = purple, p51 = gray, vRNA template = gold, and tRNA primer = red). Map rotated by 90° (right) to display end-on view of polymerase-active site. **b** 3′ tRNA$^{Lys}_3$ terminus (red) is not engaged with the polymerase-active site (see **c** for comparison to a structure with the primer terminus engaged). The polymerase-active site, primer grip, and 3′ tRNA$^{Lys}_3$ are displayed within their corresponding EM density. The β-OG molecule is hidden for clarity (Supplementary Fig. 4a). **c** The miniRTIC (purple/gold/red) is captured in an inactive polymerase conformation. The miniRTIC 3′ tRNA terminus is displaced away from the active site and the finger and thumb subdomains are hyperextended (~6 and ~5 Å, respectively) compared to an active polymerase RT–dsDNA complex (1RTD [https://doi.org/10.2210/pdb1rtd/pdb], pink). Models are aligned on palm domain backbone residues ("Methods").

**Table 1 Cryo-EM data collection, refinement, and validation statistics.**

| | apo RT/dsRNA (EMD-22899 [https://www.ebi.ac.uk/pdbe/entry/emdb/EMD-22899], PDB 7KJV [https://doi.org/10.2210/pdb7kjv/pdb]) | RT/dsRNA with nevirapine (EMD-22901 [https://www.ebi.ac.uk/pdbe/entry/emdb/EMD-22901], PDB 7KJX [https://doi.org/10.2210/pdb7kjx/pdb]) | RT/dsRNA with efavirenz (EMD-22900 [https://www.ebi.ac.uk/pdbe/entry/emdb/EMD-22900], PDB 7KJW [https://doi.org/10.2210/pdb7kjw/pdb]) |
|---|---|---|---|
| *Data collection and processing* | | | |
| Magnification (nominal) | ×165,000 | ×165,000 | ×165,000 |
| Voltage (kV) | 300 | 300 | 300 |
| Electron exposure (e⁻/Å²) | 100–107 | 100–107 | 100–107 |
| Defocus range (μm) | −1.0 to −2.5 | −1.0 to −2.5 | −1.0 to −2.5 |
| Pixel size (Å) | 0.82 | 0.82 | 0.82 |
| Symmetry imposed | C1 | C1 | C1 |
| Initial particle images (no.) | 4,315,173 | 2,656,613 | 3,072,304 |
| Final particle images (no.) | 1,344,402 | 1,155,315 | 1,123,557 |
| Map resolution (Å) | 2.8 | 3.1 | 2.9 |
| FSC threshold | 0.143 | 0.143 | 0.143 |
| Map local resolution range (Å) | 2.8–6 | 3.0–12 | 2.8–24 |
| *Refinement* | | | |
| Initial model used (PDB code) | 3V81 | 7KJV | 7KJV |
| Model resolution (Å) | 2.9 | 3.2 | 3.0 |
| FSC threshold | 0.5 | 0.5 | 0.5 |
| Map sharpening *B* factor (Å²) | −100 | −120 | −100 |
| Model composition | | | |
| Non-hydrogen atoms | 8361 | 8589 | 8561 |
| Protein residues | 941 | 948 | 949 |
| Nucleotides | 43 | 43 | 43 |
| Ligands | 1 Mg²⁺, 1 BOG, 1 G47 | 1 Mg²⁺, 1 NVP, 1 G47 | 1 Mg²⁺, 1 EFZ, 1 G47 |
| *B* factors (Å², min/max/avg) | | | |
| Protein | 33.57/119.89/57.43 | 27.28/118.20/55.39 | 23.42/91.50/40.00 |
| Nucleotides | 20.00/134.63/112.39 | 20.00/177.09/139.27 | 20.00/115.26/90.39 |
| Ligand | 55.13/116.30/90.23 | 25.36/142.80/92.94 | 20.00/115.29/71.26 |
| R.m.s. deviations | | | |
| Bond lengths (Å) | 0.003 | 0.005 | 0.007 |
| Bond angles (°) | 0.47 | 1.035 | 0.911 |
| *Validation* | | | |
| MolProbity score | 1.55 | 1.35 | 1.36 |
| Clashscore | 6.21 | 5.20 | 4.64 |
| EMRinger score | 3.76 | 2.85 | 4.14 |
| Poor rotamers (%) | 0.26 | 0.12 | 0.87 |
| Ramachandran plot | | | |
| Favored (%) | 96.66 | 97.65 | 97.33 |
| Allowed (%) | 3.34 | 2.35 | 2.56 |
| Disallowed (%) | 0.00 | 0.00 | 0.11 |
| RNA Outliers | | | |
| Backbone (no., %) | 3, 6.97 | 2, 4.65 | 2, 4.65 |
| Pucker (no., %) | 0, 0.00 | 0, 0.00 | 0, 0.00 |

sidechains within both subunits were resolved[12,13]. Several regions of disorder, mostly confined to the p66 finger subdomain, were not modeled and any unresolved sidechains were truncated (see "Methods"). Globally, the RT protein backbone is nearly identical to that of previous RTIC structures (p66 backbone r.m.s. d. = 1.2 Å (6WAZ [https://doi.org/10.2210/pdb6waz/pdb]) and 0.8 Å (6HAK [https://doi.org/10.2210/pdb6hak/pdb]), p51 backbone r.m.s.d. = 0.96 Å (6WAZ [https://doi.org/10.2210/pdb6waz/pdb]) and 0.93 Å (6HAK [https://doi.org/10.2210/pdb6hak/pdb])). Density consistent with the size of an octyl-β-glucoside molecule (β-OG), a cryo-EM freezing additive, was observed adjacent to the NNRTI-binding pocket (Supplementary Fig. 4a) and density for a Mg²⁺ ion was clearly observed in the RNase H-active site (Supplementary Fig. 4c).

Within the RT-binding cleft, density for the RNA phosphate backbone, sugars, and bases can be readily distinguished, allowing de novo modeling of the extended PBS helix. The extended PBS helix adopts an A-form helical conformation that spans the entire RT-binding cleft from the polymerase-active site to the RNase H domain (Fig. 2a and Supplementary Fig. 4k). The four nucleotides of the vRNA-template overhang, and the peripheral tRNA stem, including the bulged A and GNRA tetraloop, exhibit poor density and were not modeled (Supplementary Fig. 4j, k). Density corresponding to the cross-link between p66 C258 and tRNA G71 is visible along the minor groove of the PBS helix (Supplementary Fig. 4b).

**miniRTIC polymerase-active site adopts an inactive conformation**. The miniRTIC, despite displaying activity in vitro, is captured in an inactive state unsuitable for nucleotide incorporation, as previously observed in lower-resolution studies of the initiation complex. The tRNA 3′ primer terminus is displaced ~5 Å from the canonical elongation primer site (P-site), away from the catalytic triad (D110, D185, and D186), and resides in the P′-site. This displacement is accompanied by an ~3-Å shift of the primer grip toward the thumb and away from the polymerase-active-site residues. Collectively, these movements shift the primer terminus away from the palm subdomain, disrupting protein–nucleic acid contacts responsible for coordinating the polymerization reaction. In addition, the p66 thumb opens by ~6 Å and adopts a hyperextended conformation reminiscent of RT–DNA–NNRTI structures as reported in past analyses of lower-resolution RTIC structures[12–14]. The tip of the fingers also hyperextends by ~5 Å (Fig. 2c) to facilitate binding to the wide A-form vRNA–tRNA template–primer complex. Despite this hyperextension, D76, located at the base of the fingers, appears to make weak polar contact with the vRNA template strand (Fig. 3a) in a manner similar to that observed in prior RT–DNA–DNA elongation complexes[14,15,20].

**The RTIC features a sparse protein–RNA contact landscape**. The rigid A-form dsRNA PBS helix in the initiation complex is

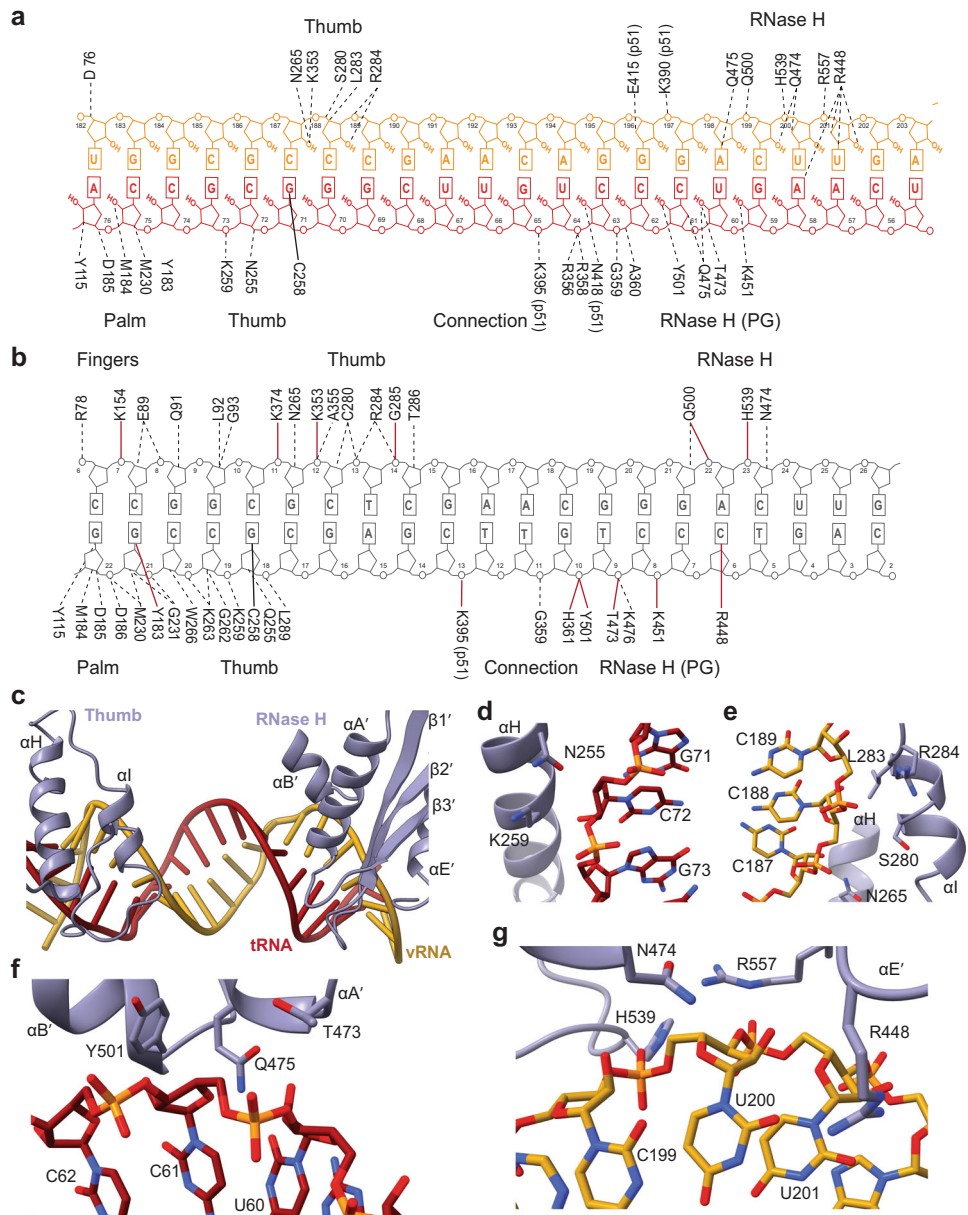

**Fig. 3 RNA–protein contact landscape in the RTIC. a** RT–dsRNA contact landscape of the miniRTIC. Secondary structure of the extended PBS helix (vRNA template = gold, tRNA primer = red), and associated contacts with protein residues. **b** RT–dsDNA contact landscape of an elongation complex (Huang et al.[20]) reveals more contacts compared to the RTIC. Primer strand on bottom, template strand on top. Hydrogen bonds specified in solid red. Solid black line indicates cross-link. RNase H primer grip abbreviated as "RNase H (PG)". **c** Top-down view of miniRTIC p66 subunit and the extended PBS helix. Thumb and RNase H domain residues of the p66 subunit (purple) contact the minor groove of the PBS helix (vRNA template = gold, tRNA primer = red). **d** View of p66 thumb contacts with the tRNA primer backbone. Residues N255 and K259 on αH helix interact with the backbone of nucleotides (nts) C72 and G73. **e** View of p66 thumb contacts with the vRNA template backbone. Residues N265 on αH helix and residues L283 and R284 on αI helix contact the backbone of nts C187, C188, and C189. **f** View of RNase H domain primer grip contacts with the tRNA backbone. Residues Y501, Q475, and T473 interact with nts U60 and C61. **g** View of residues adjacent to RNase H-active site contacting the vRNA template. Residues N474, R557, R448, and H539 interact with nts C199, U200, and U201. Nucleotides U200 and U201 are a part of the extended PBS helix.

distinct from the typical dsDNA and DNA–RNA substrates that reside in the RT-binding cleft during elongation. Globally, RT must undergo several large movements in order to accommodate the PBS helix. The previously described hyperextension of the thumb domain and shifting of the primer grip accommodates the wider A-form helix, which is raised away from the palm and connection subdomains. This movement results in disruption of crucial palm interactions, including the catalytic YMMD motif, with the primer strand[12–14]. In addition, the wider dsRNA PBS

helix makes additional contacts with the connection domain of the p51 subunit (N418, E415, and K390) compared to RT bound to a dsDNA substrate (Fig. 3a, b). Unlike prior low-resolution structural studies, which were limited to providing global descriptions and backbone deviations, the cryo-EM maps determined here allow for the accurate modeling of the specific protein–RNA contacts responsible for stabilizing the RTIC.

The majority of protein–RNA contacts within the RTIC are located in two distinct regions that are one turn apart along the

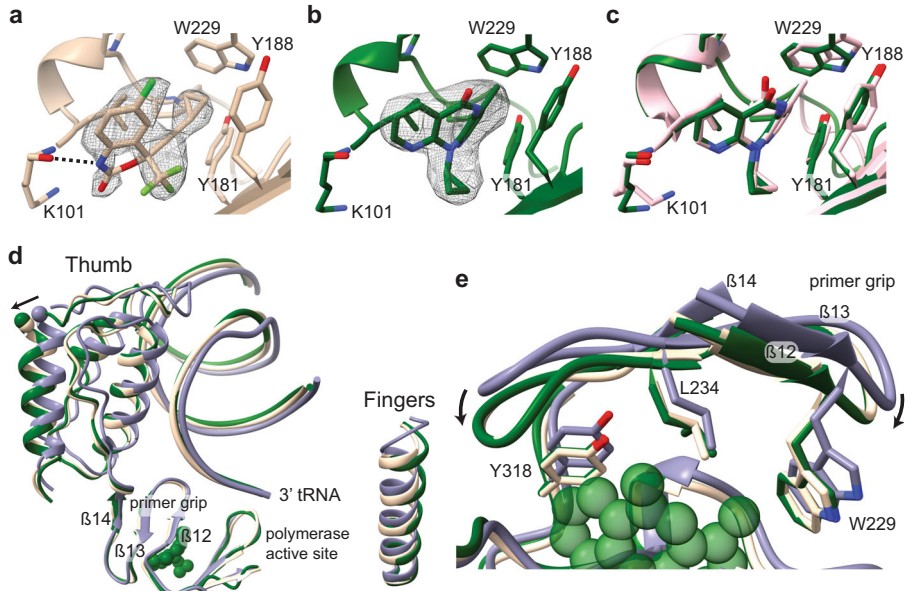

**Fig. 4 NNRTI inhibition of the RTIC. a** miniRTIC–EFZ NNRTI-binding pocket. EFZ N1 atom forms a hydrogen bond (black dashed line) with the oxygen atom on the backbone of K101. Corresponding EFZ density in mesh from 2.9-Å EM map. Y181 and Y188 flip toward the polymerase-active site to accommodate the drug. **b** miniRTIC–NVP NNRTI-binding pocket. Corresponding NVP density in mesh from 3.1-Å EM map. Similar to **a**, Y181 and Y188 flip to accommodate the drug. **c** Superposed binding pocket of miniRTIC–NVP (green) and RT–DNA–NVP (pink, 3V81 [https://doi.org/10.2210/pdb3v81/pdb]) after alignment on palm domain backbone residues ("Methods"). **d** Superposed apo-miniRTIC (purple), miniRTIC–EFZ (beige), and miniRTIC–NVP (green) after alignment on palm domain backbone residues ("Methods"). NNRTI binding locks the primer grip, causing the thumb to hyperextend by ~2 Å. **e** NNRTI binding causes compaction of the primer grip by ~2 Å to assist in the hydrophobic packing of the drug. The primer grips of both miniRTIC–NVP (green) and miniRTIC–EFZ (beige) exhibit similar compaction motions in reference to apo-miniRTIC (purple).

minor groove of the PBS helix. In the first region, two thumb subdomain alpha helices, α-H and α-I (residues 254–266 and 276–284) reside along the minor groove of the PBS helix between tRNA$^{Lys}_3$ nts 71–73 and vRNA nts 186–187 (Fig. 3c). The tRNA primer backbone contacts residues N255 and K259 on α-H (Fig. 3d), while the vRNA template backbone contacts residues L283 and R284 on α-I and N265 on α-H (Fig. 3e). Notably, interactions between the RNA and β-18, which forms a charged groove with α-I during elongation, are absent, as are interactions with the N-terminal portion of α-H that typically contact the primer strand. As such, contacts in this region are sparse compared to those found in RT–dsDNA and RT–DNA–RNA complexes (Fig. 3a, b).

A second set of contacts between RT and the PBS helix arise approximately one RNA helical turn away toward the RNase H domain (Fig. 3a and Supplementary Fig. 5d). In a similar fashion to RT–DNA complexes, RNase H domain helices α-A′ and α-B′ (residues 474–488 and 500–509) and sheets β-1′ and β-2′ (residues 438–447 and 452–459) are positioned along the minor groove of the PBS helix (tRNA nts 61–55 and vRNA nts 197–204) (Fig. 3c). RNase H primer grip residues G359, A360, K451, Y501, Q475, and T473 interact with the tRNA primer backbone (Fig. 3f), and residues adjacent to the RNase H-active site (N474, R557, H539, and R557) interact with the vRNA template strand, including two of the four base pairs of the extended PBS (Fig. 3g). Similar to past RT–DNA structures (Fig. 3g), R448 interacts along the minor groove above tRNA nt 58 and vRNA nt 200. In addition, there is density consistent with an Mg$^{2+}$ ion bound by D443, D549, and the phosphate of vRNA nt 200 (Supplementary Fig. 4c). While the vRNA backbone resides near the active site, the RNA is not engaged, which is consistent with the presence of a vRNA–tRNA template–primer substrate in the RNase H domain[21]. Compared to other RT–DNA nucleic acid complexes, several previously described RNase H-nucleic acid close-contact

hydrogen bonds are missing (H361, Y501, T473, K476, K451, R448, Q500, and H539).

**Structure of the miniRTIC–NVP and miniRTIC–EFZ.** We next sought to investigate the mechanism of NNRTI action on reverse transcription initiation. Cross-linked, minimal RTICs were purified and imaged by cryo-EM using similar methods described above; however, prior to freezing complexes, we incubated the miniRTIC with either 1 mM NVP or 200 μM EFZ for an hour at room temperature to ensure drug saturation. The structures of the miniRTIC–NVP and miniRTIC–EFZ complexes were resolved at overall resolutions of 3.1 and 2.9 Å, respectively (Supplementary Figs. 2e–l, 3c, d, f, g 6e, f and Table 1). Similar to the apo-miniRTIC model, for the two different NNRTI-bound complexes, the high-quality density of each map allowed for de novo modeling of RT and the extended PBS helix. Again, peripheral RNA elements and flexible regions of RT were left unmodeled due to poorly resolved cryo-EM density.

We were able to identify immediately prominent density for both drugs within the NNRTI-binding pocket, validating our cryo-EM approach for RT–vRNA–tRNA$^{Lys}_3$–NNRTI structural studies (Fig. 4a, b). Globally, the miniRTIC–NVP and miniRTIC–EFZ protein backbones are nearly identical to that of the apo-miniRTIC structure (p66 backbone r.m.s.d. = 0.91 Å (miniRTIC–NVP) and 0.8 Å (miniRTIC–EFZ), p51 backbone r. m.s.d. = 0.41 Å (miniRTIC–NVP) and 0.34 Å (miniRTIC–EFZ)). Inspection of the protein–RNA contact landscape revealed no significant deviations from the apo-miniRTIC, suggesting that NNRTIs may not significantly alter the already poor affinity of RT for vRNA–tRNA$^{Lys}_3$ template–primer substrates (Supplementary Fig. 5b, c). The drug conformations and drug–RT contacts within the NNRTI-binding pockets closely resemble those of past structural characterizations of RT-NNRTI complexes (Fig. 4c and Supplementary Fig. 6i). Both NNRTIs make

similar van der Waals contacts with the hydrophobic residues within the pocket (Supplementary Fig. 6b, g, h). EFZ makes an additional interaction, with its N1 atom forming a hydrogen bond with the oxygen atom on the backbone of K101, a previously observed interaction in RT–EFZ complexes representative of elongation[16,22] (Fig. 4a).

**NNRTIs exacerbate an inactive conformation of the RTIC.** Comparison of the apo-miniRTIC with both miniRTIC–NVP and EFZ structures revealed several key structural differences induced by NNRTI binding. Consistent with past structural studies of NNRTI-drug complexes, Y181 and Y188 rotate toward the active site to accommodate the drug in the binding pocket (Fig. 4a, b and Supplementary Fig. 6a). Presumably, repositioning of W229 in the binding pocket must also occur; however, in our apo-miniRTIC structure, the bound β-OG appears to have shifted the position of W299, obscuring the extent of its movement upon drug binding (Supplementary Fig. 4a). NNRTI binding favors a shift of the primer grip (β12–β13 hairpin) toward the p51 subunit, a motion that appears to assist in the hydrophobic packing of NVP and EFZ with F227, W229, and L234. In addition, the β13–β14 loop, which contains P236, moves toward the bound inhibitors by ~2 Å (Fig. 4e and Supplementary Movie 1). Together, the movements of β12–β13–β14 serve to compress the partially open NNRTI-binding pocket of the RTIC around the inhibitors, resulting in a conformation similar to that observed in past RT-NNRTI structures[15,22–24].

Despite the minimal rearrangement of the NNRTI-binding pocket, the thumb subdomains of both miniRTIC–NNRTI structures hyperextend by an additional ~2 Å upon NNRTI binding (Fig. 4d). The additional thumb hyperextension and primer grip compression slightly reposition the tRNA 3′ terminus ~0.5 Å closer to the catalytic triad; however, the terminal 3′ OH remains ~5 Å displaced from the P-site location required for catalysis (Fig. 4d). Correspondingly, this repositioning causes the vRNA template strand to lift ~0.5 Å away from the base of the fingers. Superposition of the palm subdomains of the apo- and miniRTIC–NNRTI structures reveals that the RNase H-active site has also shifted slightly by ~2 Å, a less significant movement that occurs upon NNRTI binding to an RT–dsDNA complex[15] (Supplementary Fig. 6k). Collectively, these observations indicate that NNRTI binding exacerbates the hyperextended thumb conformation of the RTIC, which further favors a 'resting' inactive state by reducing the ability of RT to sample an active-state conformation.

**NNRTIs exacerbate RT pausing and inhibit initiation.** Given the structural effects of NNRTI binding on the RTIC, we performed additional biochemical RT extension assays on an un-cross-linked RTIC formed using full-length tRNA$^{Lys}_3$ primer (76 nts) and a 101-nucleotide vRNA template construct. This construct, previously employed during low-resolution cryo-EM studies, includes all peripheral RNA elements required for efficient initiation[11–13]. RT extension reactions were initiated in the presence or absence of 3 μM NVP or 3 μM EFZ and quenched at time points of 5, 10, and 30 min. Extended tRNA$^{Lys}_3$ products were resolved at the single-nucleotide resolution on a denaturing Urea-PAGE sequencing gel (Fig. 5a). Reverse transcription, in the absence of drug, is efficient and displays the distinct pausing pattern previously reported[2–6]. After 30 min, the +3 pause is nearly alleviated and ~73% of the tRNA primer is extended (Fig. 5b). The addition of 3 μM NVP or 3 μM EFZ considerably slows the reaction. At 5 min, tRNA primer usage decreases 1.5- and 3.2-fold, for NVP and EFZ, respectively, indicating strong inhibition of reverse transcription initiation (Fig. 5b).

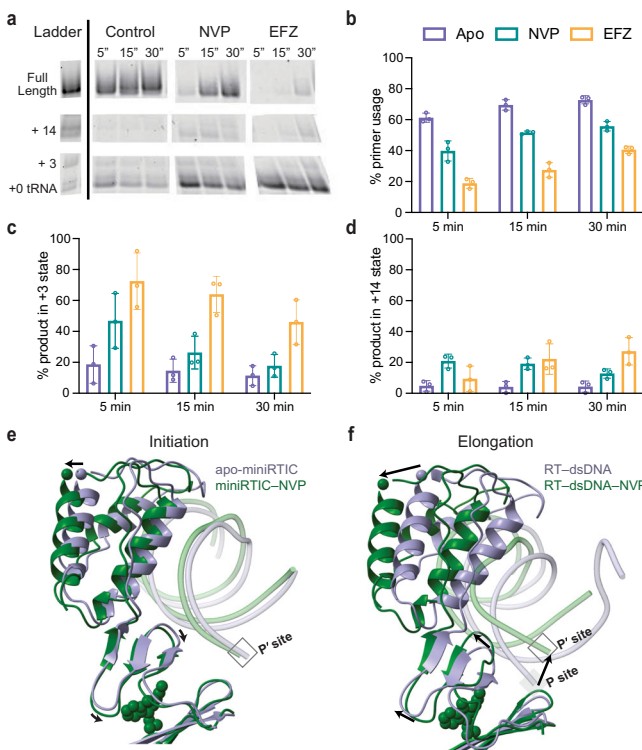

**Fig. 5 NNRTIs exacerbate RT pausing during initiation. a** Representative sequencing gel of results from reverse transcription assays performed with and without NNRTIs. Identified and labeled bands of +0 tRNA, +3, +14, and full-length transcripts. Gels are representative of n = 3 independent replicates. **b** Addition of EFZ or NVP decreases the primer usage at all time points (apo-miniRTIC=purple, miniRTIC–EFZ=yellow, and miniRTIC–NVP=green). **c** The +3 pause is exacerbated by both NVP and to a greater extent by EFZ. **d** Addition of NVP or EFZ exacerbates the +14 pause. The effect is more pronounced for EFZ at later time points due to greatly inhibited polymerization. Values for **b**, **c**, and **d** are mean ± s.d. (n = 3 independent replicates per timepoint). **e** In the RTIC, the apo structure assumes a conformation with hyperextended thumb, open-binding pocket, and 3′ terminus in the P′ site. Upon NVP binding, there is additional hyperextension of the thumb and compaction of the primer grip by ~2 Å, but otherwise there is minimal conformational change upon NNRTI binding due to the open conformation of the NNRTI binding pocket (Fig. 4e). **f** In RT–dsDNA elongation complexes, NVP binding requires and causes larger conformational changes: the primer grip must lift toward the thumb by ~4 Å to open the NNRTI binding pocket, the 3′ terminus is displaced by ~5 Å into the P′ site, and the thumb hyperextends by ~7 Å. Superpositions in **e** and **f** are aligned along palm domain backbone residues ("Methods").

Notably, NVP appears to exacerbate the +3 and +14 pausing by 2.5- and 4.4-fold, indicating that a greater amount of partially extended primer is trapped in these stalled states (Fig. 5c, d). Stalling in the presence of NVP is partially alleviated during later time points, yet remains more prominent than in the absence of the drug. In agreement with its higher efficacy, EFZ causes more severe inhibition of reverse transcription. We detected minimal amounts of full-length transcripts after 5 min, and by 30 min, only ~40% of the primer was extended (Fig. 5b). Similar to NVP, EFZ also increases the amount of extended primer stalled at the +3 and +14 states (5 min: 3.8- and twofold increase, respectively) (Fig. 5c, d). Unlike NVP, EFZ increases stalling over the course of 30 min (30 min: 4.0- and 6.3-fold increase), likely due to RT's diminished ability to extend past regions of vRNA secondary structure. Thus, both NVP and EFZ have inhibitory effects on

reverse transcription initiation and early elongation, consistent with prior studies[19].

## Discussion

Here, we provide a high-resolution view of the core of the HIV-1 RTIC prior to nucleotide incorporation, demonstrating how the dsRNA PBS helix is recognized by RT. The structure of the extended PBS helix within the binding cleft of the miniRTIC is unaffected by the lack of the peripheral, flexible RNA elements previously observed in the full-length RTIC (vRNA helix 1, vRNA helix 2, connecting loop, and extended tRNA$^{Lys}_3$ helix)[12,13]. The shortened tRNA helix of the miniRTIC, which coaxially stacks with the extended PBS helix in the context of a larger RTIC, exhibits poor density, reflecting disorder at the periphery of the complex (Supplementary Fig. 4j). Consistent with previous observations, the p66 finger–thumb clamp adopts a hyper-extended open conformation compared to elongation RT–DNA complexes, and the 3′ primer terminus is displaced from the canonical P-site location within the polymerase-active site[12–14] (Fig. 2c). Moreover, structures of the RTIC with NNRTIs bound, as well as biochemical data highlighting NNRTI inhibition of RTIC polymerization, reveal how these ligands stabilize the hyperextended conformations of the thumb to exacerbate the inactive conformation of the RTIC.

Critically, our observation that RT adopts a unique mode of engagement with the dsRNA PBS helix, coupled with the ensemble of past RT–nucleic acid structures, suggests that RT is able to alter its conformation subtly to accommodate the various helical substrates it encounters throughout the reverse transcription process[25]. During early initiation, the PBS helix within the RTIC forces RT to adopt a hyperextended conformation to accommodate the rigid wide structure of an A-form dsRNA helix. As reverse transcription progresses into elongation, the DNA–RNA and dsDNA substrates transition into a mixture of more flexible A- and B-like geometries (Supplementary Fig. 6j). The malleability of dsDNA and DNA–RNA elongation substrates has been linked to RT processivity and fast polymerization rates[26,27]. By contrast, the nonprocessive and slow nature of initiation has been associated with the rigidity and width of A-form RNA helices[2–6]. Thus, we hypothesize that the structural properties of nucleic acid substrates within the RT-binding cleft are an important factor for how efficiently the enzyme can perform its enzymatic activities during RT initiation.

Similarly, our structural findings—of disengagement of the tRNA primer terminus from the active site, and of an unusual polymerase conformation during early initiation–suggest a high degree of plasticity during the catalytic cycles of initiation. These observations suggest that, during each catalytic cycle, (i) the primer grip must return to its canonical conformation and the tRNA 3′ primer terminus must translate into the polymerase-active site, and (ii) the RTIC must reside in this conformation long enough to bind an incoming dNTP and perform the critical chemistry step of polymerization. Critically, this second step must occur prior to RT dissociation or shifting of the tRNA primer terminus back into the P′ site. Alternatively, the distributive nature of initiation suggests that this second step could also be achieved by the dissociation and subsequent rebinding of RT in a polymerase competent conformation.

In our structures of early initiation, the RTIC features a sparse protein–nucleic acid contact landscape that likely underlies the poor affinity of RT for the vRNA–tRNA$^{Lys}_3$ template–primer complex. Hyperextension of the thumb and inherent rigidity of the A-form PBS helix causes the displacement of the tRNA 3′ primer terminus away from the palm domain and the catalytic triad. Compared to elongation complexes, this conformation, which is unique to the RTIC, features fewer contacts between the PBS helix and the thumb and palm subdomains. The dearth of stabilizing interactions between the RNA and the RT polymerase domain helps to explain the slow kinetics of polymerization and lack of active-state RTIC structures. The RNase H domain, despite featuring a more substantial contact landscape during initiation, is slightly deviated away from the PBS vRNA template strand compared to its position in RT–dsDNA and RT–DNA–RNA complexes, likely reducing the strength of electrostatic and van der Waals interactions in this region. Yet additional contacts within the RNase H domain may stabilize the extended PBS helix and, in turn, the coaxially stacked tRNA within the full RTIC. Recent work on the +3 extended RTIC suggested that RNase H contacts can cause the RTIC to adopt off-pathway conformations, further highlighting their importance during initiation[13]. While the miniRTIC features primarily van der Waals and electrostatic RT–dsRNA contacts (Fig. 3a), we note that our minimal complex lacks the vRNA helices located immediately adjacent to the RT finger domain that could promote an increased network of interactions. Further, we cannot rule out the existence of direct contacts between RT and the structured vRNA template or the tRNA primer beyond the RTIC core presented here. Such interactions have been suggested in a recent cryo-EM study, but it lacked the necessary resolution to identify any specific RT–RNA contacts[13].

NNRTIs exploit the conformational landscape of the RTIC to stabilize the complex in an inactive conformation, which exacerbates the nonprocessive nature of initiation. In previously reported studies of ternary RT–dsDNA–NNRTI complexes, NNRTI binding caused destabilization of the complex by hyperextension of the fingers and thumb subdomains, dramatic repositioning of the primer grip, and shifting of the 3′ primer terminus into an unproductive P′-site location[15,28] (Fig. 5f). Superposition of previously determined RTIC complexes onto RT–dsDNA–NNRTI ternary complexes showed marked similarities, with hyperextended thumb and fingers, an expanded NNRTI binding pocket, and the 3′ primer terminus positioned at the P′ site[12–14]. A similar, open conformation of the RTIC has suggested a mechanism for its susceptibility to inhibition by NNRTIs. Our structural data reveal that binding of NNRTIs to the RTIC causes the thumb domain to hyperextend by an additional ~2 Å compared to the apo-RTIC, shifting the RTIC further into a conformation unsuitable for polymerization (Fig. 5e). This additional thumb hyperextension appears to be caused by the compaction of the NNRTI-binding pocket: the apical end of the primer grip (β13–β14 hairpin) shifts ~2 Å toward the base of the NNRTI-binding pocket, assisting in hydrophobic packing of the drugs with the greasy residues within the pocket (Supplementary Movie 1). Due to an inherently semi-open NNRTI-binding pocket, the RTIC requires only minimal conformational rearrangements to accommodate NNRTI binding. Our biochemical data suggest that by stabilizing the complex in an inactive conformation, NNRTIs exacerbate the distinct pausing features of initiation and greatly reduce polymerization rates during initiation. Our combined structural and biochemical results suggest a mechanism by which NNRTIs readily bind a susceptible, open conformation of the RTIC and then shift the complex into an inactive conformation incapable of effectively navigating the structured vRNA template.

Structural characterization of the HIV-1 RTIC has remained elusive for decades due to the inherent dynamic and unstable nature of RT–dsRNA complexes. While recent studies have overcome this barrier to provide low-resolution descriptions, our approach here allows for the detailed characterization of the contact landscape between RT and the PBS helix and for the direct visualization of NNRTIs within the NNRTI binding pocket. Our study supports the notion that NNRTIs stabilize the RT in a

distorted inactive conformation, worsening the ability of the RTIC to efficiently catalyze polymerization. Notably, fewer conformational changes within the RTIC are required for NNRTI association compared to elongation complexes, potentially highlighting a susceptibility of initiation to inhibition. Our approach provides a platform for the detailed structural characterization and further design of potential antiviral compounds that specifically target the RT–dsRNA complex of early reverse transcription, a pressing need in the backdrop of increasing drug resistance. Further work in the context of larger RNAs that span the entire initiation complex will be required to delineate the interplay of a local and global RNA structure, conformational flexibility, and inhibition of RT by drugs.

## Methods

**Cryo-EM sample preparation.** HIV-1 vRNA constructs were prepared by in vitro transcription with T7 RNA polymerase[12,29,30]. Transcripts were denatured in formamide, purified using 10 or 20% sequencing urea PAGE, and gel-extracted using 0.3 M ammonium acetate. Following ethanol precipitation, the RNA was dissolved in 10 mM Bis-Tris propane, pH 7.0, 10 mM NaCl, and stored at −20 °C until use. The cross-linkable tRNA$^{Lys}_3$ construct was purchased from TriLink Biotechnologies. The cross-linkable tRNA$^{Lys}_3$ construct was chemically synthesized, PAGE-purified, and analyzed by denaturing PAGE and mass spectrometry. During synthesis, an N2-cystamine-2′-deoxyguanosine was placed at the 71$^{st}$ position for cross-linking purposes.

Minimal vRNA–tRNA complexes were formed by mixing vRNA at 2.2 μM and tRNA at 2 μM in 10 mM Bis-Tris propane, pH 7.0, 10 mM NaCl. The mixture was heated to 95 °C and slowly cooled to room temperature. The presence of an annealed bimolecular product was analyzed by native PAGE (Supplementary Fig. 1d). The biomolecular vRNA–tRNA was annealed immediately before preparing the cross-linked miniRTIC.

HIV-1 RT was expressed in *Escherichia coli* strain BL21(DE3)[12]. Two expression vectors, one containing p66 (ampicillin resistance) and the other containing p51 (kanamycin resistance) were constructed. The C-terminus of p66 contains an unstructured linker and a cleavable six-histidine tag. A cysteine mutation for cross-linking was introduced into alpha-helix H of p66 (Q258C). The protein used in this study also had the C280S mutation, introduced in prior structural work, and the E478Q mutation, introduced to eliminate RNase H activity as RT has been shown to cleave dsRNA when stalled for long periods[31]. Cells were grown in LB medium at 37 °C until an optical density at 600 nm reached a value of 0.6. Cells were induced with the addition of 1 mM isopropyl-β-D-1-thiogalactopyranoside and grown overnight at 19 °C. Cell pellets were lysed through sonication (Lysis buffer: 100 mM NaCl, 50 mM Tris-HCl (pH 8.0), 5 mM β-ME, 5% glycerol, and 1 mM imidazole), and the enzyme was purified by gravity Ni-nitrilotriacetic acid (Ni-NTA) affinity chromatography (Wash buffer: 300 mM NaCl, 50 mM Tris-HCl (pH 8.0), 5 mM β-ME, 5% glycerol, and 10 mM imidazole. Elution buffer: 300 mM NaCl, 50 mM Tris-HCl (pH 8.0), 20 mM β-ME, 5% glycerol, and 100 mM imidazole) followed by an initial size-exclusion chromatography (SEC) step using Superdex 200 (26/600) (SEC buffer: 300 mM NaCl, 50 mM Tris-HCl (pH 8.0), and 5 mM β-ME). The six-histidine tag was cleaved by thrombin digestion overnight. The cleaved protein was reapplied to Ni-NTA column to remove protein with uncleaved six-histidine tag (SEC buffer above). This was followed by an additional final size-exclusion chromatography polishing step (SEC buffer above). The protein was stored at 4 °C in 300 mM NaCl, 50 mM Tris (pH 8.0), and 5 mM β-ME prior to use.

The miniRTIC was prepared by mixing RT and vRNA–tRNA complex at 2 μM and 1 μM, respectively, in a final buffer containing 75 mM NaCl, 25 mM KCl, 6 mM MgCl$_2$, and 50 mM Tris, pH 8.0. The mixture was allowed to cross-link overnight at room temperature. The complex was purified by anion-exchange chromatography with a linear gradient (75 mM to 1 M NaCl). This was followed by a size-exclusion chromatography step to remove any higher-molecular-weight aggregates. The purity and homogeneity of the final complex were assessed by SDS-PAGE (under nonreducing conditions, Supplementary Fig. 1c), native page (Supplementary Fig. 1d), and size-exclusion chromatography (Fig. 1c). The purified miniRTIC was stored in a buffer of 10 mM Tris, pH 8.0, and 75 mM NaCl overnight at 4 °C.

**Cryo-EM data acquisition.** Immediately prior to grid freezing, 0.2% (w/v) beta-octyl glucoside (β-OG) and 6 mM MgCl$_2$ were added to apo-miniRTIC. NVP and EFZ were purchased from Sigma in powder form. The ligands were dissolved in DMSO to generate stock solutions of NVP at 82.6 mM and EFZ at 47.5 mM concentrations. The ligands were subsequently diluted in 10 mM Tris, pH 8.0, and 75 mM NaCl, prior to addition to complexes. For the drug-bound complexes, miniRTIC–NVP and miniRTIC–EFZ were separately incubated with 1 mM NVP and 200 μM EFZ for 1 h prior to the addition of β-OG and MgCl$_2$.

Samples of 30 μM apo-miniRTIC, 60 μM miniRTIC–NVP, and 60 μM miniRTIC–EFZ were applied to glow-discharged Quantifoil grids (R 0.6/1 100

Holey Gold Supports + 2 nm C Grids: Au 200 mesh) and subsequently vitrified using an FEI Vitrobot (100% humidity, 22 C, blot force 3, 2 s, 1.5-s blot time, respectively).

Frozen hydrated samples were imaged on an FEI Titan Krios (300 kV) with a Gatan K2 Summit direct detection camera and energy filter in counting mode with 200-ms exposure per frame. The dose rate was 8.0 electrons per pixel per second and forty frames per micrograph were collected at a magnification of ×165,000 (corresponding to 0.82 Å per pixel at the specimen level). The total dose was 100–107 electrons per Å$^2$ across all three datasets. In total 12,186, 12,675, and 12,402 micrographs were collected at defocus values ranging from −1.0 to −2.5 μm for the apo-miniRTIC, miniRTIC–NVP, and miniRTIC–EFZ, respectively (Table 1). Data collection was performed using SerialEM[32] and Gatan DigitalMicrograph.

**Cryo-EM processing.** The movie frames were motion-corrected and dose-weighted by MotionCor2 and CTF parameters were estimated by GCTF[33,34]. Cryo-EM data were processed using Relion 3.0 and cryoSPARC[35–37]. In total, 4,315,173, 2,656,613, and 3,072,304 particle projections were semiautomatically picked in RELION 3.0 from the apo-miniRTIC, the miniRTIC–NVP, and the miniRTIC–EFZ datasets, respectively. The particle projections were extracted, downscaled four times, imported, and sorted through subsequent rounds of reference-free 2D classification in cryoSPARC. In total, 1,587,090, 1,155,315, and 1,129,740 particle projections belonging to classes with well-defined protein and RNA features were selected and re-extracted with two-time binning for further processing. A 3D ab initio model was generated using cryoSPARC based on the selected 2D classes, used for 3D classification in RELION 3.0. In total, 1,344,402, 1,155,315, and 1,123,557 particle projections were selected and re-extracted without binning for subsequent 3D refinement from the apo-miniRTIC, the miniRTIC–NVP, and the miniRTIC–EFZ datasets, respectively. The 3D reconstructions were refined to a resolution of 2.8, 3.1, and 2.9 Å, respectively, and sharpened in RELION 3.0 (Supplementary Fig. 2). Additional attempts to further classify the three datasets did not improve the final resolution of the reconstructions and typically resulted in less well-ordered density for the RT finger subdomain and the vRNA–tRNA PBS helix (Supplementary Fig. 3a).

**Model building and refinement.** For the apo-minRTIC, a crystal structure of RT, with nucleic acid removed, was used as an initial starting model (PDB: 3V81 [https://doi.org/10.2210/pdb3v81/pdb]) by rigid docking in Chimera. Regions of major structural differences in RT were de novo modeled in Coot[36], as was the entirety of the PBS helix. Atomic coordinates were refined by iteratively performing Phenix real-space refinement[38,39] followed by manual inspection and correction in Coot. Secondary structure restraints were enabled, and a map weight was used to avoid overfitting. In regions of low map quality, side-chain atoms were truncated to alanine or residues were removed entirely. Modeling of the miniRTIC–NVP and miniRTIC–EFZ were performed as described above with the exception that the apo-miniRTIC (PDB: 7KJV [https://doi.org/10.2210/pdb7kjv/pdb]) was used as the starting model. Ligand structures were downloaded from the RCSB PDB Ligand Expo or fetched using Coot's monomer library. Initial docking of the EFZ and NVP ligands was performed by aligning the ligands to prior RT–NNRTI structures and then fit into the density using Coot. The β-OG ligand was docked into its respective density and initially fit using Coot. Restraints for all ligands were generated by eLBOW in Phenix prior to real-space refinement. Final models and maps were validated using Phenix comprehensive validation, including MolProbity[40] and EMRinger[41] analysis. Nucleic acid helicity was analyzed with 3DNA[42] (Supplementary Fig. 6j). Molecular graphics and analyses performed with Coot, Phenix, and ChimeraX[38,39,43,44].

Apo-miniRTIC modeling: p66 subunit (chain A) modeled residues 4–62, 71–133, 142–557, p51 subunit (chain B) modeled residues 6–218, 231–356, 363–427, vRNA (chain C) modeled residues 182–203 and tRNA (chain D) modeled residues 55–76. p66 subunit truncated side-chain residues 22, 27–29, 31–32, 34–36, 38–40, 42–43, 46, 49–50, 53, 71, 73, 78–79, 89–92, 131, 142–143, 151, 169, 173–174, 177, 195, 197, 220, 248, 250, 286–287, 289, 297, 311, 334, 344, 357, 396, 404, 413, 432, 449, 512, 516. p51 truncated side-chain residues 11, 13, 22, 65–68, 79, 86, 113, 138, 169, 173, 174, 194, 203, 212, 214–215, 238, 240–241, 250, 275, 297–298, 315, 334, 404, 424.

miniRTIC–EFZ modeling: p66 subunit modeled residues 4–63, 70–134, 141–558, p51 subunit modeled residues 5–216, 231–357, 362–428, vRNA modeled residues 182–203 and tRNA modeled residues 55–76. p66 subunit truncated side-chain residues 28, 32, 36, 40, 43, 50, 53, 70, 79, 86, 92, 117, 143, 177, 191, 194, 218, 220, 286–287, 290, 324, 334, 428, 448–449, 471, 557–558. p51 subunit truncated side-chain residues 5–6, 11, 22, 66–67, 86, 162, 166, 174, 214, 240, 250, 297–298, 315, 334, 356, 388, 428.

miniRTIC–NVP modeling: p66 subunit modeled residues 2–64, 70–134, 141–557, p51 subunit modeled residues 6–214, 233–426, vRNA modeled residues 182–203 and tRNA modeled residues 55–76. p66 subunit truncated side-chain residues 29, 36, 40, 43, 53, 64, 70, 92, 110, 194, 255, 286–287, 297, 308, 324, 344, 428, 461, 514. p51 truncated side-chain residues 6, 11, 66–67, 69, 86, 89, 113, 177, 194–195, 250, 297, 298, 357, 358.

**Alignments**. Due to chain breaks in modeling, r.m.s.d. calculations comparing apo-miniRTIC, 6WAZ, and 6HAK for p66 used residues 4–21, 35–50, 77–89, 93–126, 142–190, 198–215, 253–430, 470–545 (CA, C, O, N). r.m.s.d. calculations comparing apo-miniRTIC, 6WAZ, and 6HAK for p51 used residues 6–93, 95–166, 175–210, 231–240, 250–284, 293–350, 363–425 (CA, C, O, N). r.m.s.d calculations comparing apo-miniRTIC, miniRTIC–NVP, and miniRTIC–EFZ for p66 used residues 4–21, 35–50, 77–89, 93–126, 142–190, 198–215, 253–430, 470–545 (CA, C, O, N). r.m.s.d. calculations comparing apo-miniRTIC, miniRTIC–NVP, and miniRTIC–EFZ for p51 using residues 6–16, 20–211, 236–300, 320–355, 365–416 (CA, C, O, N).

**Protein–RNA contact map analysis**. Contacts were first identified with a < 5-Å cutoff distance between RT and the vRNA–tRNA. After identification, these contacts were inspected and curated (see Fig. 4a and Supplementary Fig. 5). For Supplementary Fig. 5d, the total number of atomic "contacts" (all non-hydrogen protein atoms ≤5 Å from all non-hydrogen nucleotide atoms) for each individual nucleotide were identified and counted using Pymol.

**Time-course assays**. The cross-linked miniRTIC, RT, and vRNA–tRNA used in $^{32}$P activity assays were prepared as described above. miniRTIC (50 nM) was preincubated in 50 mM Tris-HCl, pH 8.0, 50 mM KCl, and 6 mM MgCl$_2$ in a 37 °C water bath for 5 min. For drug conditions, nevirapine (50 nM) and efavirenz (50 nM) were separately incubated with miniRTIC (50 nM) under the same conditions. Free vRNA–tRNA (50 nM) and RT (250 nM) was also preincubated under the same conditions. Incorporation reactions were initiated by adding a mixture of α-$^{32}$P-dCTP (40 nM) and dCTP (50 µM). Reactions were quenched at various time points between 5 s and 4 h with the addition of EDTA and SDS loading buffer. The reaction sets for each condition (free, x-link, x-link + nevirapine, x-link + efavirenz) were run on a 12% SDS-PAGE gel, dried, and exposed for 18 h on a phosphoimager screen (Molecular Dynamics) and each gel was individually scanned with a Storm 860 (Molecular Dynamics)[12,13]. Bands were quantified using ImageQuant. The intensity was normalized to the highest intensity for the individual time-course assays after background subtraction (set to 1). The miniRTIC and free vRNA–tRNA with RT were repeated five times. Nevirapine and efavirenz conditions were each repeated three times. Plotting and curve fitting was performed using GraphPad Prism8.

**Reverse transcriptase assay**. vRNA–tRNA complexes were purified as described above using a tRNA primer with a 5′ cyanine-3 dye (Cy3) label[12,13]. Reactions were preincubated at 37 °C for 5 min in 50 mM Tris-HCl, pH 8.0, 50 mM KCl, 6 mM MgCl$_2$, and 4 mM β-ME at vRNA–tRNA concentration of 200 nM and RT at a concentration of 3 µM. For drug conditions, preincubation occurred under the same conditions and with the addition of 3 µM efavirenz or 3 µM nevirapine. Reactions were initiated by the addition of a dNTP mixture concentration of 100 µM. Reactions were performed in triplicate and quenched at 5-, 10-, and 30 min with EDTA and 2X formamide dye. For the unextended control sample, preincubation conditions were the same, followed by quenching and then the addition of 100 µM dNTP mixture. Samples were denatured and heated for 5 min at 95 °C and loaded on an 8.5% polyacrylamide gel that was prerun for 2 h at 100 W. Samples were run for 3 h at 120 W before imaging for Cy3 fluorescence with a Typhoon Trio (Amersham Biosciences). Unextended primer bands were quantified using GE ImageQuant. Percent primer extension was normalized to control and plotting/curve fitting was performed using GraphPad Prism8[13].

**Reporting summary**. Further information on experimental design is available in the Nature Research Reporting Summary linked to this paper.

## Data availability

Cryo-EM density maps have been deposited in the EMDB with accession codes EMD-22899 (apo-miniRTIC), EMD-22901 (miniRTIC–NVP), and EMD-22900 (miniRTIC–EFZ). Atomic coordinates have been deposited in the PDB with accession codes 7KJV (apo-miniRTIC), 7KJX (miniRTIC–NVP), and 7KJW (miniRTIC–EFZ). All other data are available from the corresponding author upon reasonable request. Source data are provided with this paper.

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

## Acknowledgements

We thank N.R. Latorraca and J.D. Puglisi for critical reading and comments on the paper, N. Moriarty for assistance with modeling the cross-link, and N.R. Latorraca for assistance with molecular rendering software and computations. This work was supported by National Institutes of Health grant AI150464 to E.V.P., T32-GM008294 (Molecular Biophysics Training Program) to K.P.L., T32-HG000044 (Stanford Genome Training Program) to B.H., National Science Foundation Graduate Research Fellowship Program (DGE-1656518) to L.N.J., and the Knut and Alice Wallenberg Foundation postdoctoral scholarship to J.Z. We thank Stanford University and the Stanford Research Computing Center for providing the Sherlock cluster resources. Some of this work was performed on a Thermo Fisher Scientific Krios at the Stanford-SLAC Cryo-EM Facilities, supported by Stanford University, SLAC, and the National Institutes of Health S10 Instrumentation Programs. Molecular graphics and analyses performed with UCSF ChimeraX, developed by the Resource for Biocomputing, Visualization, and Informatics at the University of California, San Francisco, with support from National Institutes of Health R01-GM129325 and the Office of Cyber Infrastructure and Computational Biology, National Institute of Allergy and Infectious Diseases.

## Author contributions

K.P.L. and E.V.P. conceptualized the minimal RTIC construct design for high-resolution studies. K.P.L. designed the purification scheme. B.H. and K.P.L. performed all vRNA and RT sample preparations. B.H. prepared all samples for cryo-EM data acquisition. B.H., L.N.J., J.Z., E.M., and D.-H.C. prepared grids for cryo-EM imaging. J.Z., E.M., and D.-H.C. acquired preliminary and experimental cryo-EM data. Z.F. and J.Z. obtained final 3D reconstructions. B.H. performed all α-$^{32}$P-dCTP incorporation assays. B.H performed all reverse transcription assays. B.H. and K.P.L. performed RTIC model building and refinement. B.H., K.P.L., and E.V.P. interpreted the data. B.H., K.P.L., and E.V.P. wrote the paper. E.V.P. supervised all experiments, data analysis, and writing.

## Competing interests

The authors declare no competing interests.
