## [Peer Review File · Nature Communications]

REVIEWER COMMENTS

Reviewer #1 (Remarks to the Author):

This manuscript by Ha et al., describes the cryo-EM structures of a minimal HIV reverse transcriptase initiation complex (RTIC) and its complex with two HIV inhibitors, nevirapine and efavirenz. Built on the previous structural studies of the full RTIC at medium resolution, a minimal vRNA-tRNA duplex was designed to reduce conformation flexibility and improve the cryo-EM resolution. Taking advantage of the improved resolution compared to the previous larger RTIC assembly, these structures allowed detailed visualization of the interactions between the vRNA-tRNA duplex and HIV RT in an initiation state and interactions of this core RTIC with the two inhibitors. Interestingly, the minimal RTIC adopts an extended conformation with the substrate 3' terminus in an inactive conformation (away from the RT active site). The binding of the two non-nucleoside inhibitors exacerbates this open inactive conformation. The work provides great insight into HIV RT initiation and is of great significance to the HIV and RT fields. The associated biochemical studies nicely support the observations made in these structures.

While I find the work interesting, there are a number of technical points that need to be addressed to strengthen the conclusions.

1. Regarding the cryo-EM data processing shown in Sup. Fig. 2, each reconstruction is resulted from 3D refinement of 1.1-1.3 million particles, which is unusually high (in many cases, one can achieve this with less than 100k particles). The data may have been insufficiently classified. This is reflected in Fig. 2a. Secondary structural features do not show as clearly as one would expect and the map has spiky features on the periphery. Given the observed conformational heterogeneity in parts of the complex, more 3D classification would be needed to dissect this further and likely improve the resolutions of not only these flexible parts but also the whole complex.
2. The model vs map FSC validation is missing in the EM validation. Also nucleic acid geometry validation should also be included in Table 1.
3. Sup. Figures 4d and g show the RNA duplex with a very broken backbone density, indicating heterogeneity present. Perhaps more classification suggested above would help resolving this.
4. In the experiments determining the incorporation rate of the next nucleotide (Fig. 1d and Sup. Fig. 1d), the time courses for different sets of complexes have different saturated signals, especially between the uncrosslinked and crosslinked complexes. Were these experiments done in parallel? If so, is there any explanation for the discrepancy observed.
5. In the discussions, the authors noted the lack of the vRNA helices which could promote additional RNA-RT interactions. How does the incorporation rate of the full RTIC compare to the minimal RTIC? Does the presence of the additional vRNA helices change the incorporation rate?
6. It is intriguing that despite the incorporation of the incoming nucleotide, the 3' end of the substrate is very far away from the active site. What would be the explanation for this?
7. For general audience not familiar with the structure, it would be very helpful to have a good figure showing a cartoon overview of the structure with domains and subunits labelled and coloured and an associated domain schematics. Figure 2a contains only the EM map, which is very difficult to relate to when during the remainder of the paper. Also an indication of where the drug binding site is in this overview structure would be very useful.

Minor points/typos:

Figure 1a. It is very difficult to see the RNA part that was greyed out in the schematics. If it is to be shown, it would be helpful to find a different way to present this (perhaps making it visible and the RNA construct used with a grey box underneath). In the figure legend, there are missing spaces at

“50 nM”. Same for all other uses of concentrations in the paper.

Figure 3b. The schematic describes a dsDNA-RT contacts. However, the sugar has 2' OH groups. Likely a copy-and-paste error?

Figure 3d. K259 was described to contact the RNA. What is the distance between this residue and the RNA? In this figure the amine group of the lysine seems to point away from the backbone.

Figure 4d and e. Grey and light purple are quite similar in shade. It may be helpful to use a darker purple shade.

Line 255, referred to Fig. 4a. Should this be Fig. 4d?

Reviewer #2 (Remarks to the Author):

Ha, Larsen, Puglisi et al. present in this manuscript three structures corresponding to minimal reverse transcriptase initiation complexes (RTICs), with an improvement of >1 Å resolution in comparison to previous RTIC structures (by removing the elements protruding outside the RT nucleic acid binding cleft). In concordance to earlier structures suggesting ligandability of the RTIC for NNRTIs, two of the current RTIC structures present NNRTI drugs bound. This manuscript establishes thus a benchmark for structure-based drug discovery targeting RTICs and provides a more detailed look to the distorted conformation of the RTICs that might explain the much slower catalysis in comparison to RT elongation complexes. It is an excellent work that deserves publication, well written, with very detailed and good figures, albeit some minor changes should be done.

I will list next some comments/suggestions, as well as some minor edits that are needed for publication, in order of appearance:

- I would suggest a slight addition in the title: “High-resolution view of HIV-1 Reverse Transcriptase initiation complexes and inhibition by NNRTI drugs”. This is to avoid confusing reads.
- Page 2, line 54: whenever the two strands of the nucleic acid are not identified (e.g., vRNA or tRNA), I would specify the convention, which seems to be “primer–template”, with something like “RNA-RNA primer–template, from now on in this order”.
- Page 3, line 66: “x-ray crystallographic” should be changed to “X-ray crystallographic”.
- Page 3, line 78: “(x-ray)” should be changed to “(X-ray crystallography)”.
- Page 3, line 80: “drug action” should be changed to “drug binding”. While it is true that observing the detailed binding of a drug can unveil the drug action, this is not always the case.
- Page 4, line 92: you may comment in between the two sentences that the two first RTIC structures determined by cryo-EM and X-ray crystallography showed a partially open NNRTI binding pocket not observed before in any RT structure without a bound NNRTI, and that your lab previously observed that NVP inhibited an RTIC complex. These data, in addition to the comments you already wrote, were making the experiments you present here very sensible and appealing to do.
- Page 4, line 103: change end of the sentence to “miniRTIC-NNRTI complexes with NVP and EFV (3.1 and 2.9 Å resolution, respectively)”. Otherwise, although implied, it is not clear that the quantities here are referring to resolution.
- Page 5, line 121: this is related to fig. 1c: I suppose that the absorbance data is at 280 nm. Please, specify somewhere.
- Page 6, line 146: A frontal view of beta-OG would be better for assessing its fitting to the electron density map.
- Page 6, line 149: “de novo” should be in italics.
- Page 7, lines 153-154: “(Supplementary Fig. 4j)” should be changed to “(Supplementary Fig. 4j and 4k)”. 4k illustrates the 2D info seen in 4j and may help the reader.
- Page 7, line 155: Given that the density is there for the cross-linking (Supplementary Fig. 4b), why the cross-linking was not modeled? The modified G coordinates and restraints are already in the

PDB: <https://www.rcsb.org/ligand/MRG> .

- Page 7, line 157: “in vitro” should be in italics. There are several other instances where this has to be done, as well for other terms as “in silico” or “de novo”. Also, “r.m.s.d” should be changed to “r.m.s.d.”.

- Page 7, lines 167-169: What do you define as a contact between RT and RNA? Can you elaborate about the nature of the D76-RNA contact? It seems that in RT-RNA/DNA-dATP complex (PDB ID 4PQU) the 2'-OH of the second nucleotide overhang (-2) interacts with the side chain of D76. This residue is a conserved residue at the base of the functionally critical $\beta 3$ - $\beta 4$ sheet that is critical for template binding. The interaction with DNA seems like a weak polar interaction in PDB ID 1rtd. Also in PDB IDs 3v6d and 6hak (the last a lower-resolution RTIC). Also, see interesting paper related to D76 and R78 role in template binding:

<https://www.jbc.org/content/274/39/27666.full.pdf>

All in all, my take is that, while the tip of the fingers opens, the base of the fingers is roughly in the same conformation as in elongation complexes. In comparing all these structures, it would seem that this part of the base of the RT fingers can handle in slightly different ways the binding of different nucleic acid binding partners. It would be interesting to compare the similarities and differences, as seen in the figure 2b-2e in here: <https://www.pnas.org/content/116/15/7308/tab-figures-data> . Finally, in the large movements described in the last sentences of page 7 and start of page 8, I think that the hyperextension of the tip of the fingers is also needed for proper binding of the dsRNA, so I would add it.

- Page 9, line 215: is the different concentration used for both NNRTIs due to solubility issues with EFZ? Also, I believe that the solvent(s) in which the NNRTIs were dissolved is not indicated in the methods.

- Page 10, line 231: in connection to the comment on “page 7, lines 167-169”, I wonder whether the addition of the contact of R78, which as reasoned before is usually interacting with D76 in elongation complexes, will provide a slightly better binding than in the apo complex, hence contributing to longer pausing (?). Comparison of this residue pair between apo and NNRTI-bound RTICs could be helpful in this regard.

- Comments on subsection “NNRTIs lock the RTIC in an inactive conformation”: figs. 4a and 4b depict electron density for the bound NNRTIs from a top view, as well as 4c depicts molecular interactions. Again, a frontal view would be more helpful for viewing it. As a reference, figs. 3 and 4 in here: <https://www.sciencedirect.com/science/article/pii/S0304416514001342> , may illustrate my point better. Also, superpositions with RT-NNRTI and RT-nucleic acid-NNRTI can be included in similar orientations.

Additionally, is there any evidence of NNRTIs locking the RTIC inactive conformation? I believe that the biochemical evidences are solid (i.e., longer pausing), but in terms of the structure I do not see it so clear. It is true that there is additional hyperextension, but one would assume that locking the RTIC in a conformation would suppose less flexibility (motion). According to the B-factors, there seems to be no difference between apo and NNRTI-bound complexes. It is well established that the meaning of B factors is quite clear in crystallography, but it is more ambiguous in cryo-EM ([https://www.cell.com/structure/fulltext/S0969-2126\(17\)30246-0?_returnURL=https%3A%2F%2Flinkinghub.elsevier.com%2Fretrieve%2Fpii%2FS0969212617302460%3Fshowall=true](https://www.cell.com/structure/fulltext/S0969-2126(17)30246-0?_returnURL=https%3A%2F%2Flinkinghub.elsevier.com%2Fretrieve%2Fpii%2FS0969212617302460%3Fshowall=true)). Assuming the B-factors are reliable, would not this mean that the difference in motion between the three structures is negligible? Also, the local resolution seems to be slightly worse in the NNRTI bound datasets (specially with NVP). Could this be a sign that these complexes could be a bit more dynamic than the apo complex?

- Page 13, lines 302-305: related to the previous comment, I would rephrase the sentence:

“Moreover,

structures of the RTIC with NNRTIs bound reveal how these ligands may stabilize and exacerbate...”. The structure-activity relationships drawn here (but not the structures by themselves) prove, in my opinion, that NNRTIs may lock the RTIC in an inactive conformation. In fact, I think the sentences in page 16 lines 371-376 really summarize it perfectly.

- Page 13, lines 306-309: I would rephrase it, because it is already inferred from the ensemble of many RT/nucleic acid structures in the PDB that “RT is able to subtly alter its conformation to accommodate the various helical substrates it encounters throughout the reverse transcription process.”. Indeed, this is reviewed graphically in fig. 2 in here: <https://www.sciencedirect.com/science/article/pii/S0959440X1930137X> .
- Page 18, line 416-418: “...the enzyme was purified by gravity Ni-nitrilotriacetic acid (Ni-NTA) affinity chromatography using Superdex 200 (26/600).” I think this may be corrected, as the latter is a size exclusion chromatography.
- I think that the authors should at least indicate in the methods section how the restraints of the ligands were generated. Also, as overfitting of the ligand can be a concern in cryo-EM and the validation reports may show some signs of it (a bond length (or angle) with $|Z_j| > 2$ is considered an outlier worth inspection, and this is the case for some bonds of both NVP and EFZ in the structures), adopting a pipeline such as GemSpot (<https://www.sciencedirect.com/science/article/pii/S0969212620301398>) could be helpful in this regard.

Reviewer #3 (Remarks to the Author):

The manuscript by Ha et al describes structures of a HIV reverse transcriptase mini-initiation complex of that includes the viral polymerase, a 26nt viral RNA template and a 39 nt truncated tRNA-lys3 in apo form, and bound to a non-nucleotide inhibitor, nevirapine or efavirenz to near atomic resolutions. This is the continuation of the group’s previous successful structural characterization of HIV replication initiation. HIV reverse transcription begins with recruiting of human tRNA-lys3 as a primer, this unique feature remains an attractive drug target for antiviral reagents design. Structural information of the process is of eminent biological significance. Recently, two HIV RT initiation complex structures have been determined, a cryo-EM structure with an intact tRNA-lys3 from the authors’ group and a crystal structure of Arnold’s group. Interestingly, the cryo-EM structure adopts an inactive conformation whereas the crystal structure represents an active conformation. While the reported structures have achieved commendable improvement in resolution, the mini-initiation complexes, with or without inhibitors, are all apparently in inactive conformations, similarly to the previous cryo-EM structure. Therefore, the knowledge gained in understanding of HIV RT initiation from the current manuscript is limited. The inactive conformation of mini-initiation complex also diminishes certainty of conclusion that the drug bound HIV RT yield inactive conformation.

It would be helpful if the authors provide hypothesis and explanation on the cause and biological meaning of the inactive conformation, and what step in the reverse transcription initiation process the inactive conformation represents. It is indeed puzzling that the vRNA/tRNA construct is kinetically active but inactive in structures. This reviewer noticed that the conditions for kinetic analyses has lower ionic strength than the structural studies, which may contribute to the inactive location of the primer strand.

January 30, 2021

We first would like to thank the 3 reviewers for their careful and constructive critiques, while all recognizing the importance of our high-resolution structures for understanding reverse transcription initiation. To address the reviewer concerns, we have reanalyzed our cryoEM data, added additional figures and greatly clarified our methods, results, presentation and terminology where confusing. We took the reviewer comments to heart, and have addressed them in detail below, as outlined in the underlined text. We hope that the revised and improved manuscript is now acceptable for publication in Nature Communications.

Reviewer #1 (Remarks to the Author):

This manuscript by Ha et al., describes the cryo-EM structures of a minimal HIV reverse transcriptase initiation complex (RTIC) and its complex with two HIV inhibitors, nevirapine and efavirenz. Built on the previous structural studies of the full RTIC at medium resolution, a minimal vRNA-tRNA duplex was designed to reduce conformation flexibility and improve the cryo-EM resolution. Taking advantage of the improved resolution compared to the previous larger RTIC assembly, these structures allowed detailed visualization of the interactions between the vRNA-tRNA duplex and HIV RT in an initiation state and interactions of this core RTIC with the two inhibitors. Interestingly, the minimal RTIC adopts an extended conformation with the substrate 3' terminus in an inactive conformation (away from the RT active site). The binding of the two non-nucleoside inhibitors exacerbates this open inactive conformation. The work provides great insight into HIV RT initiation and is of great significance to the HIV and RT fields. The associated biochemical studies nicely support the observations made in these structures.

While I find the work interesting, there are a number of technical points that need to be addressed to strengthen the conclusions.

1. Regarding the cryo-EM data processing shown in Sup. Fig. 2, each reconstruction is resulted from 3D refinement of 1.1-1.3 million particles, which is unusually high (in many cases, one can achieve this with less than 100k particles). The data may have been insufficiently classified. This is reflected in

Fig. 2a. Secondary structural features do not show as clearly as one would expect and the map has spiky features on the periphery. Given the observed conformational heterogeneity in parts of the complex, more 3D classification would be needed to dissect this further and likely improve the resolutions of not only these flexible parts but also the whole complex.

The reviewer correctly observed that the final number of particles for each reconstruction is large compared to many other reconstructions of a similar resolution. Unfortunately, we have found that our past reconstructions of HIV-1 RTICs have all required a considerable number of particles to achieve modest improvements in resolution. The improvement from our original 4.5 Å reconstruction to a 4.1 Å reconstruction required three times as many particles (130k to 400k) despite extensive, focused classification efforts (Larsen et al. 2018 and 2020). Based on these past experiences, we expected that the modifications to our RNA construct design would need to be coupled with an even larger dataset to generate reconstructions better than 3.5 Å.

As the reviewer keenly noted, despite this large dataset, several peripheral, flexible regions of our maps exhibit relatively poor structural features for the reported resolution (namely the RT p66 fingers subdomain and many solvent exposed sidechains). **To address the concern that particles were not sufficiently classified, we performed additional focused/masked classification on all three miniRTIC datasets as now shown in Sup. Fig. 3a.** Despite our best efforts, which include focused/masked classification, the subclasses within each particle set consistently resulted in final reconstructions at a lower resolution with poorer density in flexible regions. While we have succeeded in identifying subclasses, these new reconstructions often fail to consistently resolve the RNA at a nucleotide level and feature worse density for the fingers domain. Notably, the major differences within many of these new maps appears to reside at the junction between the extended PBS helix and tRNA, which are disordered in our final maps and not a major topic of discussion.

A simplified example of our additional classification efforts can be seen in Sup. Fig. 3a. While we would welcome improvements in our final maps, we believe that the maps represent the current best reconstruction that we can provide and are detailed enough for the analysis contained within the manuscript.

2. The model vs map FSC validation is missing in the EM validation. Also nucleic acid geometry validation should also be included in Table 1.

Model vs Map FSC validation has now been added to Sup. Fig. 2. Nucleic acid geometry validation, including backbone and sugar pucker outliers, has been added to Table 1.

3. *Sup. Figures 4d and g show the RNA duplex with a very broken backbone density, indicating heterogeneity present. Perhaps more classification suggested above would help resolving this.*

It has been our experience, with the HIV-1 RTIC and other RNA cryo-EM datasets, that the density corresponding to the RNA backbone phosphates falls off much more quickly than density corresponding to nucleobases and protein (Larsen et al. 2018 and 2020). This phenomenon often

requires that the protein and RNA aspects of these complexes are viewed and modeled at different contour thresholds. As such, the broken backbone density previously depicted in Sup. Fig. 4 is a consequence of the chosen density contour threshold rather than due to an additional level of underlying heterogeneity. **To alleviate this discrepancy, we have now included additional figure panels of the RNA duplex density in Sup. Fig. 4k that display the maps at several different contour thresholds.** Regular density for the phosphates is now visible along the RNA backbone at the low and medium threshold contours. Unsurprisingly, density for the RNA is much weaker in regions that lack RT–RNA contacts, likely due to the absence of stabilizing interactions. As discussed above, our attempts to further classify the data have failed to increase the quality of the RNA duplex density, which often results in reconstructions featuring poorly resolved nucleobases.

4. In the experiments determining the incorporation rate of the next nucleotide (Fig. 1d and Sup. Fig. 1d), the time courses for different sets of complexes have different saturated signals, especially between the uncrosslinked and crosslinked complexes. Were these experiments done in parallel? If so, is there any explanation for the discrepancy observed.

We agree with the reviewer that the raw gel images (Sup. Fig. 1d) appear to show varying degrees of saturated signal (i.e. band intensity) between the uncrosslinked and crosslinked complexes. While the experiments were performed in parallel (using the same preparations of reverse transcriptase and/or miniRTIC), the quenched reactions for each time series (uncrosslinked, crosslinked, crosslinked w/NNRTI) were run on separate SDS-PAGE gels, dried and exposed on a phosphoscreen, then individually scanned for P32 signal. As such, the apparent band intensity differences are a consequence of the varying degrees of background signal, which is subtracted during analysis, for each individually prepared and scanned gel. Our prior published studies of the crosslinked HIV-1 RTIC (Larsen et. al. 2018, ED Fig 1d,e) presented similar P32-dNTP incorporation experiments parallelized within the same gel for accurate, direct comparisons. These experiments show that the crosslinked and uncrosslinked HIV-1 RTICs saturate at nearly identical levels and therefore incorporate similar amounts of dNTP. We would like to note the possibility that the crosslinked complexes in presence of NNRTI may require additional extremely long timepoints to reach true saturation due to the slow kinetics. However, after >4 hours the signal no longer increases, likely due to competing undesirable processes such as complex degradation and aggregation (high temperature and lack of reducing agents required to minimize crosslink dissociation).

The apparent discrepancy with Fig. 1d thus appears to be visual and related to the curve fitting performed for the crosslinked complexes w/NNRTI. In the prior manuscript, the curves ended at the last data point rather than extending the entirety of the x-axis. We have updated the figure to extend the curves past the last point. Despite this update, the curves fit to the NNRTI data suggest that additional longer time points may be required for true saturation (~8 hrs). However, as previously mentioned above, we have found that reactions past 4 hours do not result in greater signal likely due to undesired competing processes. Nonetheless, we believe the rates extracted from the current data accurately convey the manuscript's point that NNRTIs have a dramatic inhibitory effect on the HIV-1 RTIC.

5. In the discussions, the authors noted the lack of the vRNA helices which could promote additional RNA-RT interactions. How does the incorporation rate of the full RTIC compare to the minimal RTIC? Does the presence of the additional vRNA helices change the incorporation rate?

The incorporation rate of the first templated dNTP is similar between the full RTIC and the minimal RTIC (Page 5, line 126-127) (Larsen et. al 2018), suggesting that the additional vRNA helices do not affect the incorporation of the first dNTP. We have not measured rates of incorporation past the first templated dNTP and suspect that later rates would be affected due to the lack of secondary structure within the template strand. **In the revised manuscript, we have included an explicit statement that the structures in this manuscript only represent the state before the incorporation of the first dNTP during RT initiation. (Page 6, line 132 and page 13, line 300-301)**

6. It is intriguing that despite the incorporation of the incoming nucleotide, the 3' end of the substrate is very far away from the active site. What would be the explanation for this?

To date, all structures of the initiation complex solved by us (Larsen et al. 2018 and 2020) and others (Das et al. 2019) have shown the 3' primer terminus distant from the active. Our single nucleotide incorporation assays suggest that the minimal RTIC is catalytically active, however, as the reviewer mentions, we captured the complex with the 3' end shifted away from the active site using cryo-EM. Interestingly, the observed displacement of the 3' end is independent of structure determination method (cryo-EM: Larsen *et al.* 2018 and 2020 and x-ray crystallography: Das *et al.* 2019). We believe that the inherent rigidity of the A-form RNA–RNA PBS helix, combined with its weak contact with RT, contribute to RT's inability to easily bend the nucleic acid duplex and position the 3' terminus within the polymerase active site (a feature observed in all RT–DNA–DNA and RT–RNA–DNA complexes). The observed hyperextension of the thumb and fingers subdomains, which are not situated to properly grip the RNA duplex for catalysis, appears to be a prerequisite for the accommodation of the wider RNA–RNA A-form helix. Despite this displaced conformation captured in our cryo-EM experiments, our biochemical experiments show that the RTIC is capable of visiting the active conformation and incorporating an incoming dNTP (albeit slowly). Trapping of a potentially short-lived active state, which may display unique structural features, is a focus of future work.

7. For general audience not familiar with the structure, it would be very helpful to have a good figure showing a cartoon overview of the structure with domains and subunits labelled and coloured and an associated domain schematics. Figure 2a contains only the EM map, which is very difficult to relate to when during the remainder of the paper. Also an indication of where the drug binding site is in this overview structure would be very useful.

We appreciate the reviewer's recommendation to add additional background information. **We have added an additional panel, Sup. Fig. 1a**, featuring several views of RT with the subdomains and NNRTI binding pocket colored and labeled.

Minor points/typos:

Figure 1a. It is very difficult to see the RNA part that was greyed out in the schematics. If it is to be shown, it would be helpful to find a different way to present this (perhaps making it visible and the RNA construct used with a grey box underneath). In the figure legend, there are missing spaces at "50 nM". Same for all other uses of concentrations in the paper.

For increased clarity, Fig. 1a has now been updated to include the secondary structures of both the larger vRNA-tRNA (Larsen et. al. 2018) and the minimal vRNA-tRNA (boxed). The figure caption has been updated accordingly and all instances of missed spaces for concentrations have been corrected.

Figure 3b. The schematic describes a dsDNA-RT contacts. However, the sugar has 2' OH groups. Likely a copy-and-paste error?

This was indeed a copy/paste error during figure preparation. We have updated Fig. 3b with the 2' OH groups removed.

Figure 3d. K259 was described to contact the RNA. What is the distance between this residue and the RNA? In this figure the amine group of the lysine seems to point away from the backbone.

We have slightly altered the view of Fig. 3d to better depict the orientation of the K259 to avoid confusion. The previous Fig. 3d depicted the sidechain K259 projecting out from the page perpendicular to the RNA helix (similar to the contact observed in Huang *et al.* 1998). The amine group of the K259 sidechain comes within ~4.7 Å of the nearest backbone oxygen on G73 (tRNA primer) while other K259 atoms come within 4.0 Å of the tRNA primer

Figure 4d and e. Grey and light purple are quite similar in shade. It may be helpful to use a darker purple shade. Line 255, referred to Fig. 4a. Should this be Fig. 4d?

We have replaced the grey color for a light beige in models representing miniRTIC bound to EFZ to allow for greater contrast with the purple shade of the apo miniRTIC. Colors in corresponding panels have been updated. We have also fixed the error within Line 255 and now correctly refer to Fig. 4d.

Reviewer #2 (Remarks to the Author):

Ha, Larsen, Puglisi et al. present in this manuscript three structures corresponding to minimal reverse transcriptase initiation complexes (RTICs), with an improvement of >1 Å resolution in comparison to previous RTIC structures (by removing the elements protruding outside the RT nucleic acid binding cleft). In concordance to earlier structures suggesting ligandability of the RTIC for NNRTIs, two of the current RTIC structures present NNRTI drugs bound. This manuscript establishes thus a benchmark for structure-based drug discovery targeting RTICs and provides a more detailed look to the distorted conformation of the RTICs that might explain the much slower catalysis in comparison to RT elongation complexes. It is an excellent work that deserves publication, well written, with very detailed and good figures, albeit some minor changes should be done.

I will list next some comments/suggestions, as well as some minor edits that are needed for publication, in order of appearance:

- I would suggest a slight addition in the title: "High-resolution view of HIV-1 Reverse Transcriptase initiation complexes and inhibition by NNRTI drugs". This is to avoid confusing reads.

We appreciate the reviewer's suggestion and the **title has now been modified accordingly.**

- Page 2, line 54: whenever the two strands of the nucleic acid are not identified (e.g., vRNA or tRNA), I would specify the convention, which seems to be “primer–template”, with something like “RNA-RNA primer–template, from now on in this order”.

We have **replaced our rather confusing language with “vRNA–tRNA template–primer”** where appropriate.

- Page 3, line 66: “x-ray crystallographic” should be changed to “X-ray crystallographic”.
- Page 3, line 78: “(x-ray)” should be changed to “(X-ray crystallography)”.
- Page 3, line 80: “drug action” should be changed to “drug binding”. While it is true that observing the detailed binding of a drug can unveil the drug action, this is not always the case.

We have **remedied the above mistakes** in the text.

- Page 4, line 92: you may comment in between the two sentences that the two first RTIC structures determined by cryo-EM and X-ray crystallography showed a partially open NNRTI binding pocket not observed before in any RT structure without a bound NNRTI, and that your lab previously observed that NVP inhibited an RTIC complex. These data, in addition to the comments you already wrote, were making the experiments you present here very sensible and appealing to do.

To address this valid comment by the reviewer, we have added the following sentences on page 4, lines 91-96: “Notably, the recently determined low-resolution structures of the RTIC all exhibited a partially open RT NNRTI binding pocket, a feature previously absent in all RT structures lacking bound NNRTI¹⁸⁻²⁰. In addition, complex studied by cryo-EM was also shown to be inhibited by nevirapine in biochemical assays. However, whether or how NNRTIs perturb the conformation of an RTIC remains unknown.”

- Page 4, line 103: change end of the sentence to “miniRTIC-NNRTI complexes with NVP and EFV (3.1 and 2.9 Å resolution, respectively)”. Otherwise, although implied, it is not clear that the quantities here are referring to resolution.

This has been fixed in the text on page 4, line 106.

- Page 5, line 121: this is related to fig. 1c: I suppose that the absorbance data is at 280 nm. Please, specify somewhere.

The figure has been updated to now read “Absorbance at 280 nm (mAU)”.

- Page 6, line 146: A frontal view of beta-OG would be better for assessing its fitting to the electron density map.

We have replaced the image in Sup. Fig. 4a with a front view of beta-OG.

- Page 6, line 149: “de novo” should be in italics.

This has been fixed in the text on page 7, line 152.

- Page 7, lines 153-154: “(Supplementary Fig. 4j)” should be changed to “(Supplementary Fig. 4j and 4k)”. 4k illustrates the 2D info seen in 4j and may help the reader.

This has been fixed in the text on page 7, line 157.

- Page 7, line 155: Given that the density is there for the cross-linking (Supplementary Fig. 4b), why the cross-linking was not modeled? The modified G coordinates and restraints are already in the PDB: <https://www.rcsb.org/ligand/MRG> .

We agree with the reviewer that the crosslink should be modelled and we have coordinated with the Phenix developers to model and incorporate the crosslink into our complexes (named: DGK). **Our models and figures, including Sup. Fig. 4b, have been updated accordingly.** For our experiments we employed an N2-cystamine-2'-deoxyguanosine, which has a C2 tether rather than the C3 tether found in the MRG coordinates. Despite its use in obtaining x-ray crystallography structures of RT-nucleic acid complexes (Huang *et al.* 1998, Lansdon *et al.* 2010, Hung *et al.* 2019), the N2-cystamine-2'-deoxyguanosine has not yet been modeled and deposited into the PDB, for this reason it was not included in the original modeling.

- Page 7, line 157: “in vitro” should be in italics. There are several other instances where this has to be done, as well for other terms as “in silico” or “de novo”. Also, “r.m.s.d” should be changed to “r.m.s.d.”.

The above mistakes have been remedied throughout the text.

- Page 7, lines 167-169: What do you define as a contact between RT and RNA? Can you elaborate about the nature of the D76-RNA contact? It seems that in RT-RNA/DNA-dATP complex (PDB ID 4PQU) the 2'-OH of the second nucleotide overhang (–2) interacts with the side chain of D76. This

residue is a conserved residue at the base of the functionally critical β 3– β 4 sheet that is critical for template binding. The interaction with DNA seems like a weak polar interaction in PDB ID 1rtd. Also in PDB IDs 3v6d and 6hak (the last a lower-resolution RTIC). Also, see interesting paper related to D76 and R78 role in template binding: <https://www.jbc.org/content/274/39/27666.full.pdf> All in all, my take is that, while the tip of the fingers opens, the base of the fingers is roughly in the same conformation as in elongation complexes. In comparing all these structures, it would seem that this part of the base of the RT fingers can handle in slightly different ways the binding of different nucleic acid binding partners. It would be interesting to compare the similarities and differences, as seen in the figure 2b-2e in here: <https://www.pnas.org/content/116/15/7308/tab-figures-data> . Finally, in the large movements described in the last sentences of page 7 and start of page 8, I think that the hyperextension of the tip of the fingers is also needed for proper binding of the dsRNA, so I would add it.

Contacts were first identified with a <5 Å cutoff distance between RT and the vRNA–tRNA. After identification, these contacts were inspected and curated (see Fig. 4a and Sup. Fig. 5). We have now added this information into the Methods section for clarity.

We appreciate the reviewer's insightful comments and discussion regarding fingers subdomain interactions with the template strand within the RTIC. We have expanded upon the nature of the D76 contact, which as the reviewer suggests, appears to make weak polar contacts with the template strand in a similar manner to that found during elongation (1rtd and 3v6d) and in the prior lower-resolution RTIC structure (6hak). While we cannot fully rule out the existence of the 2'-OH interaction (due to our inability to model the template overhang bases), the orientation of the template strand with respect to the fingers suggests that such a contact does not form. In addition, the lack of well-resolved density for the template overhang suggests that this region is flexible and likely devoid of specific stabilizing contacts (at least in the absence of the structured vRNA helices). Resolving this region in greater detail, perhaps in the presence of the viral RNA helices (which may reduce the mobility of the template strand), will be key in deciphering the atomic details of RT's ability to engage with a wide variety of nucleic acid substrates. We also agree with the reviewer that the tip of the fingers is hyperextended to likely facilitate binding with wider dsRNA helices. **To address these points, we have altered Page 7-8, lines 171-175 to now read:**

"The tip of the fingers also hyperextends by ~5 Å (Fig. 2c) to facilitate accommodation of the wide A-form vRNA–tRNA template–primer complex within the RT binding cleft. Despite this hyperextension, D76, located at the base of the fingers, appears to make a weak polar contact with the vRNA template strand (Fig. 3a) in a manner similar to that observed in prior RT–DNA–DNA elongation complexes".

We have also performed additional contact analyses similar to the figure referred to in Das *et al.* 2019. The contact landscapes across all three miniRTIC complexes are very similar (as previously seen in Sup. Fig. 4) and many apparent differences are mostly related to regions where unresolved sidechains were truncated (e.g. C-terminus of the RNase H domain). Due to our inability to resolve the template overhang nucleotides near the fingers subdomain of our initiation complexes, the plots mostly highlight the overall sparse contact landscape (relative to elongation complexes). One notable feature is that the template strand of the NNRTI bound structures is slightly raised away from the base of the fingers and palm compared to the apo miniRTIC (~0.5-1 Å). This movement appears to be coupled with the slight movement of the 3' terminus towards the catalytic triad. **We thus slightly**

expanded our description of the differences between the apo and NNRTI bound complexes (page 12, lines 265-266). We have also incorporated these plots into Sup. Fig. 5 and include references to them in the main text when referring to the RT-nucleic acid contact landscape.

• Page 9, line 215: is the different concentration used for both NNRTIs due to solubility issues with EFZ? Also, I believe that the solvent(s) in which the NNRTIs were dissolved is not indicated in the methods.

The reviewer correctly inferred that we used a lower concentration of EFZ than NVP during complex formation due to its reduced solubility. The ligands were dissolved in DMSO to generate stock solutions of NVP at 82.6 mM and EFZ at 47.5 mM concentrations. The ligands were subsequently diluted in 10 mM Tris pH 8.0 and 75 mM NaCl, prior to addition to complexes. **These details are now included in the Methods section (page 18-19, lines 445-448).**

• Page 10, line 231: in connection to the comment on “page 7, lines 167-169”, I wonder whether the addition of the contact of R78, which as reasoned before is usually interacting with D76 in elongation complexes, will provide a slightly better binding than in the apo complex, hence contributing to longer pausing (?). Comparison of this residue pair between apo and NNRTI-bound RTICs could be helpful in this regard.

The reviewer makes an intriguing hypothesis regarding the potential importance of the R78 contact in contributing to the longer pausing effect of NNRTI's during initiation. We would like to note that the lack of the R78 contact in the apo miniRTIC contact map is likely due to its truncation during model building. While the backbone density in this region is unambiguous, several sidechains, including R78, exhibit insufficient density to properly model and orient. Interestingly, R78 exhibits well-ordered density in the EFZ map, possibly suggesting that its interaction with the vRNA template is stronger, while R78 exhibits weaker density in the NVP map. While it is tempting to link the stronger R78 density to the increased potency and enhanced stalling effect of EFZ on the RTIC, we are hesitant to do so given the close alignment of the NVP and EFZ models and the lower local resolution of the NVP map in this region. We are open to the possibility that by raising of the vRNA template strand slightly away from these residues (see the discussion above), R78 may more easily interact with the vRNA phosphate backbone and therefore contribute to the enhanced pausing effect observed in our assays. However, due to the limitations of the density in our cryo-EM maps mentioned above, we are hesitant to include such conjunction in the manuscript. We would also like to leave the reviewer with the possibility that these residues (R76 and D78) play a key role during the +3 and +5 stalling points during initiation, which likely feature the close proximity of structured template RNA within this region (and with it, additional modes of interaction).

• Comments on subsection “NNRTIs lock the RTIC in an inactive conformation”: figs. 4a and 4b depict electron density for the bound NNRTIs from a top view, as well as 4c depicts molecular interactions. Again, a frontal view would be more helpful for viewing it. As a reference, figs. 3 and 4 in here: <https://www.sciencedirect.com/science/article/pii/S0304416514001342> , may illustrate my point better. Also, superpositions with RT-NNRTI and RT-nucleic acid-NNRTI can be included in similar orientations.

We have rotated the views in Fig. 4a,b and c to display more of a frontal view for each NNRTI within the binding pocket. Additionally, we have added the requested full frontal view of each ligand in Sup. Fig. 6c,d and refer to them in the text.

Additionally, is there any evidence of NNRTIs locking the RTIC inactive conformation? I believe that the biochemical evidences are solid (i.e., longer pausing), but in terms of the structure I do not see it so clear. It is true that there is additional hyperextension, but one would assume that locking the RTIC in a conformation would suppose less flexibility (motion). According to the B-factors, there seems to be no difference between apo and NNRTI-bound complexes. It is well established that the meaning of B factors is quite clear in crystallography, but it is more ambiguous in cryo-EM

([https://www.cell.com/structure/fulltext/S0969-](https://www.cell.com/structure/fulltext/S0969-2126(17)302460?_returnURL=https%3A%2F%2Flinkinghub.elsevier.com%2Fretrieve%2Fpii%2FS0969212617302460%3Fshowall=true)

[2126\(17\)302460?_returnURL=https%3A%2F%2Flinkinghub.elsevier.com%2Fretrieve%2Fpii%2FS0969212617302460%3Fshowall=true](https://www.cell.com/structure/fulltext/S0969-2126(17)302460?_returnURL=https%3A%2F%2Flinkinghub.elsevier.com%2Fretrieve%2Fpii%2FS0969212617302460%3Fshowall=true)). Assuming the B-factors are reliable, would not this mean that the difference in motion between the three structures is negligible? Also, the local resolution seems to be slightly worse in the NNRTI bound datasets (specially with NVP).

Could this be a sign that these complexes could be a bit more dynamic than the apo complex?

We appreciate these deep comments by the reviewer regarding dynamics as inferred from these structures. **To address these valid points, we have eliminated the term “locking” from the manuscript, and emphasized hyperextended thumb conformation in the drug complexes, while avoiding a discussion of dynamics.** We employed the term “locking” to take into account both biochemical and structural data that mechanistically describe how NNRTIs bind and restrict RTICs from sampling conformations that are on-pathway for dNTP incorporation. Our intention was to comment on how the miniRTIC, which already resides in an inactive state, is further prevented from undergoing the domain closures required to accommodate the tRNA 3' OH into the polymerase active site. In fact, single-molecule studies have shown that while “NNRTI binding to RT induces opening of the fingers and thumb subdomains” (Schauer *et al.* 2014), the presence of NNRTIs does not prevent the motions arising from the inherent flexibility of these domains.

We hesitate to draw firm conclusions about the inherent dynamics/motion of any of the three complexes using the data contained within the manuscript. The less-established nature of B-factors in cryo-EM (vs crystallography) has led us to rely more on local resolution estimates, visual inspection of the cryo-EM density, and the results of 3D classification to identify regions of significant flexibility. While it is tempting to conclude that the lower local resolution in the miniRTIC-NVP fingers domain correlates with a more dynamic complex, the reconstruction has an overall lower global resolution, and we have been unable to identify subclasses featuring distinct alternative finger conformations. Because of this, we are inclined to believe that differences in motion between the three structures are negligible, and truthfully more direct measurements of dynamics either by NMR or single-molecule methods are required in the future. **We have changed the title of the subsection to “NNRTIs exacerbate an inactive conformation of the RTIC”**

Page 13, lines 302-305: related to the previous comment, I would rephrase the sentence: “Moreover, structures of the RTIC with NNRTIs bound reveal how these ligands may stabilize and exacerbate...”. The structure-activity relationships drawn here (but not the structures by themselves) prove, in my opinion, that NNRTIs may lock the RTIC in an inactive conformation. In fact, I think the sentences in page 16 lines 371-376 really summarize it perfectly.

We have rephrased the sentence on Page 13-14, line 310-314 to read: “Moreover, structures of the RTIC with NNRTIs bound, as well as biochemical data highlighting NNRTI inhibition of RTIC polymerization, reveal how these ligands stabilize the hyperextended conformations of the thumb to exacerbate the inactive conformation of the RTIC.”

• Page 13, lines 306-309: I would rephrase it, because it is already inferred from the ensemble of many RT/nucleic acid structures in the PDB that “RT is able to subtly alter its conformation to accommodate the various helical substrates it encounters throughout the reverse transcription process.”. Indeed, this is reviewed graphically in fig. 2 in here:<https://www.sciencedirect.com/science/article/pii/S0959440X1930137X> .

We have rephrased the sentence on page 14, lines 315-318 to read: “Critically, our observation that RT adopts a unique mode of engagement with the dsRNA PBS helix coupled with the ensemble of RT–nucleic acid structures suggests that RT is able to subtly alter its conformation to accommodate the various helical substrates it encounters throughout the reverse transcription process³¹”

• Page 18, line 416-418: “...the enzyme was purified by gravity Ni-nitrilotriacetic acid (Ni-NTA) affinity chromatography using Superdex 200 (26/600).”. I think this may be corrected, as the latter is a size exclusion chromatography.

We have corrected the sentence on page 18, lines 425-427 to read: “Cell pellets were lysed through sonication and the enzyme was purified by gravity Ni-nitrilotriacetic acid (Ni-NTA) affinity chromatography followed by an initial size exclusion chromatography step using Superdex 200 (26/600).”

• I think that the authors should at least indicate in the methods section how the restraints of the ligands were generated. Also, as overfitting of the ligand can be a concern in cryo-EM and the validation reports may show some signs of it (a bond length (or angle) with $jZ_j > 2$ is considered an outlier worth inspection, and this is the case for some bonds of both NVP and EFZ in the structures), adopting a pipeline such as GemSpot (<https://www.sciencedirect.com/science/article/pii/S0969212620301398>) could be helpful in this regard.

We have added details on the fitting of our ligands into the EM maps and generation of ligand restraints for refinement in the Methods section of the manuscript. In short, ligand structures were either downloaded from the RCSB PDB Ligand Expo search or fetched using Coot’s Monomer library. NVP and EFZ ligands were initially docked into their respective density using prior structural alignments as a reference. The BOG ligand was docked into its corresponding asymmetric density and initially fit using Coot. Restraints for all ligands generated by eLBOW in Phenix prior to real-space-refinement. We have re-examined and re-refined the ligands for each model to reduce the number of bond length and angle outliers. While the EFZ ligand still appears to have several outliers, we do not believe that they are substantial and represent an improvement upon many currently deposited RT-EFZ structures residing in the PDB.

In addition, we attempted to refine the ligands using Phenix/OPLS3e

(<https://www.biorxiv.org/content/10.1101/2020.07.10.198093v2.full>), which is the key refinement aspect of the full GemSpot pipeline mentioned by the reviewer. We decided to forgo full implementation of the full GemSpot pipeline due to the unambiguous fits of the NVP and EFZ ligands into the cryo-EM density based on prior structural information and the distinct density. However, we appreciate that docking and refinement of novel NNRTIs would likely greatly benefit for this pipeline. Refinement of the ligands using Phenix/OPLS3e did not result in the improvement in bond/angle outliers or in any noticeable changes to the ligand fit within the cryo-EM density. Therefore, we decided to retain the improved ligand refinements performed using Phenix and eLBOW generated restraint parameters described above.

Reviewer #3 (Remarks to the Author):

The manuscript by Ha et al describes structures of a HIV reverse transcriptase mini-initiation complex of that includes the viral polymerase, a 26nt viral RNA template and a 39 nt truncated tRNA-lys3 in apo form, and bound to a non-nucleotide inhibitor, nevirapine or efavirenz to near atomic resolutions. This is the continuation of the group's previous successful structural characterization of HIV replication initiation. HIV reverse transcription begins with recruiting of human tRNA-lys3 as a primer, this unique feature remains an attractive drug target for antiviral reagents design.

Structural information of the process is of eminent biological significance. Recently, two HIV RT initiation complex structures have been determined, a cryo-EM structure with an intact tRNA-lys3 from the authors' group and a crystal structure of Arnold's group. Interestingly, the cryo-EM structure adopts an inactive conformation whereas the crystal structure represents an active conformation.

While the reported structures have achieved commendable improvement in resolution, the mini-initiation complexes, with or without inhibitors, are all apparently in inactive conformations, similarly to the previous cryo-EM structure. Therefore, the knowledge gained in understanding of HIV RT initiation from the current manuscript is limited. The inactive conformation of mini-initiation complex also diminishes certainty of conclusion that the drug bound HIV RT yield inactive conformation.

We would like to stress to the reviewer that, to date, all experimentally determined structures of the RTIC (including the X-ray crystallography structure in Das *et al.* 2019) have been captured in an inactive state not amenable for nucleotide incorporation. Based on the prior low-resolution structures and those within this manuscript, we believe that the rigidity of the A-form vRNA-tRNA^{Lys}₃ PBS helix, and its weak contact with RT, contribute to RT's struggle to readily adopt an active conformation, as addressed above for comments to reviewer 1.

While the complex is captured in a non-active state, our activity assays show that our crosslinked complex is catalytically active in incorporating the next templated nucleotide in the absence of NNRTIs. The structures contained in the manuscript (apo and two NNRTI bound forms) provide the first high-resolution views of the RTIC and the first views of drug binding to the initiation complex. **Together, these structures provide insight into the subtle conformational changes that occur**

upon drug binding during initiation, which mirror, but are not identical to, those seen for elongation complexes. The manuscript also provides long sought near-atomic level detail on how RT contacts the unique vRNA-tRNA template primer complex. In addition, our biochemical data suggest that these drugs effectively inhibition the RTIC by exacerbating the pausing features of initiation. Due to past difficulties in obtaining high-resolution crystallographic data on the RTIC, we also believe that the miniRTIC cryo-EM platform provides researchers with a structure-based platform for future drug discovery studies.

It would be helpful if the authors provide hypothesis and explanation on the cause and biological meaning of the inactive conformation, and what step in the reverse transcription initiation process the inactive conformation represents. It is indeed puzzling that the vRNA/tRNA construct is kinetically active but inactive in structures. This reviewer noticed that the conditions for kinetic analyses has lower ionic strength than the structural studies, which may contribute to the inactive location of the primer strand.

The inactive RTIC complex represents a static state prior to catalysis as described in Larsen *et al.* 2018. This conformational state must undergo the conformational changes described in the discussion on page 14-15, lines 332-338, in order to be poised for catalysis. We acknowledge that the ionic strength in the structural studies and kinetic studies slightly differ in order to maximize the extension efficiency of RT, however we have successfully performed robust RT activity assays in buffers containing >125 mM NaCl. We would also like to note that the Arnold lab RTIC was independently determined by X-ray crystallography, rather than cryo-EM, and adopts a similar inactive state conformation (Das *et al.* 2019).

REVIEWER COMMENTS

Reviewer #1 (Remarks to the Author):

The authors have addressed my comments on the previous version satisfactorily. The work will be of great interest for the HIV and RT fields. I recommend this paper for publication in Nature Communications.

Reviewer #2 (Remarks to the Author):

The answers and edits of the authors have clarified my doubts, solved typos and inconsistencies, and provided even more depth and clarity to the paper. I want to thank the authors for addressing all the questions thoroughly, really outstanding work.
Thus, in my opinion, the manuscript is now ready for publication.

REVIEWERS' COMMENTS

Reviewer #1 (Remarks to the Author):

The authors have addressed my comments on the previous version satisfactorily. The work will be of great interest for the HIV and RT fields. I recommend this paper for publication in Nature Communications.

Reviewer #2 (Remarks to the Author):

The answers and edits of the authors have clarified my doubts, solved typos and inconsistencies, and provided even more depth and clarity to the paper. I want to thank the authors for addressing all the questions thoroughly, really outstanding work.

Thus, in my opinion, the manuscript is now ready for publication.

We thank both reviewers 1 and 2 for their kind words and recommendations. We have no other comments.